# Tip60-mediated lipin 1 acetylation and ER translocation determine triacylglycerol synthesis rate

Terytty Yang Li[1], Lintao Song[1], Yu Sun[1], Jingyi Li[1], Cong Yi[2], Sin Man Lam[3], Dijin Xu [4], Linkang Zhou[4], Xiaotong Li[1], Ying Yang[1], Chen-Song Zhang[1], Changchuan Xie[1], Xi Huang[1], Guanghou Shui[3], Shu-Yong Lin [1], Karen Reue[5] & Sheng-Cai Lin[1]

Obesity is characterized by excessive fatty acid conversion to triacylglycerols (TAGs) in adipose tissues. However, how signaling networks sense fatty acids and connect to the stimulation of lipid synthesis remains elusive. Here, we show that homozygous knock-in mice carrying a point mutation at the Ser[86] phosphorylation site of acetyltransferase Tip60 ($Tip60^{SA/SA}$) display remarkably reduced body fat mass, and $Tip60^{SA/SA}$ females fail to nurture pups to adulthood due to severely reduced milk TAGs. Mechanistically, fatty acids stimulate Tip60-dependent acetylation and endoplasmic reticulum translocation of phosphatidic acid phosphatase lipin 1 to generate diacylglycerol for TAG synthesis, which is repressed by deacetylase Sirt1. Inhibition of Tip60 activity strongly blocks fatty acid-induced TAG synthesis while Sirt1 suppression leads to increased adiposity. Genetic analysis of loss-of-function mutants in Saccharomyces cerevisiae reveals a requirement of ESA1, yeast ortholog of Tip60, in TAG accumulation. These findings uncover a conserved mechanism linking fatty acid sensing to fat synthesis.

[1] State Key Laboratory of Cellular Stress Biology, Innovation Center for Cell Signaling Network, School of Life Sciences, Xiamen University, Fujian 361102, China. [2] School of Basic Medical Sciences, Zhejiang University, Hangzhou, Zhejiang 310058, China. [3] State Key Laboratory of Molecular Developmental Biology, Institute of Genetics and Developmental Biology, Chinese Academy of Sciences, Beijing 100084, China. [4] MOE Key Laboratory of Bioinformatics and Tsinghua-Peking Center for Life Sciences, School of Life Sciences, Tsinghua University, Beijing 100084, China. [5] Department of Human Genetics, David Geffen School of Medicine at University of California, Los Angeles, CA 90095, USA. These authors contributed equally: Terytty Yang Li, Lintao Song, Yu Sun. Correspondence and requests for materials should be addressed to S.-C.L. (email: linsc@xmu.edu.cn)

In eukaryotes, triacylglycerols (TAGs) are the major energy storage molecules that allow organisms to survive during periods of nutrient deprivation[1, 2]. However, excessive TAG synthesis and storage in adipocytes may lead to obesity, a worldwide pandemic[3, 4], which is often associated with increased risk of human diseases including type 2 diabetes, cardiovascular disease and certain types of cancer[5–7]. In most mammalian tissues, TAG biosynthesis occurs primarily via the glycerol-3-phosphate pathway, through the action of conserved enzymes including glycerol-3-phosphate acyltransferases (GPATs), 1-acylglycerol-3-phosphate acyltransferases (AGPATs), phosphatidic acid phosphohydrolases (PAPs)/lipins, and diacylglycerol acyltransferases (DGATs). It is also worth mentioning that both glycerokinase (GK) and glycerol-3-phosphate dehydrogenase contribute to this pathway by providing the substrate glycerol-3-phosphate. In the small intestine, an alternative pathway involving monoacylglycerol acyltransferases (MGATs) is highly active[2].

Discovered as a mutated gene in *fld* (fatty liver dystrophic) mouse models characterized by lipodystrophy with almost complete loss of fat, *Lpin1* encodes an enzyme that catalyzes the dephosphorylation of phosphatidic acids (PAs) to form diacylglycerols (DAGs), the penultimate step in TAG synthesis[8, 9]. In addition to its function as a PAP, lipin 1 also acts as a co-regulator of DNA-bound transcription factors, such as members of the peroxisome proliferator-activated receptor (PPAR) family, sterol-response element binding protein 1 (SREBP1) and nuclear factor of activated T cells c4 (NFATc4)[8, 10, 11]. Lipin 1 is highly expressed in adipose tissue, and both the PAP activity and the nuclear localization of lipin 1 are required for the induction of adipogenesis in fibroblasts[9]. Interestingly, mice with adipocyte-specific expression of a truncated lipin 1 protein lacking PAP activity but retaining transcriptional regulatory function exhibit remarkably diminished adiposity, indicating that lipin 1-mediated PAP activity alone could play an essential role in the intermediary fat metabolism in adipocytes[12]. Different from other TAG biosynthesis enzymes including GPATs, AGPATs and DGATs, lipin 1 is not an integral membrane protein[2, 8, 9]. Therefore, it needs to translocate from cytosol to endoplasmic reticulum (ER) membranes, whereby the hydrophobic phospholipids and neutral lipids are synthesized[2] to catalyze the PAP reaction. However, how this translocalization process is achieved and regulated in response to increased availability of fatty acids remains poorly understood.

The HIV-1 Tat-interacting protein 60 kD (Tip60) is the catalytic subunit of the highly conserved NuA4 acetyltransferase complex, playing critical roles in DNA damage repair, checkpoint activation, p53-directed apoptosis, senescence and autophagy[13–15]. Although Tip60 has been mainly investigated as a transcriptional regulator, a growing body of evidence shows that Tip60 also acts as a key mediator in signal transduction pathways, such as the Ras/p38 signaling and insulin/AKT signaling, via direct interacting and acetylating non-histone proteins[15, 16]. However, the physiological functions of Tip60, especially in metabolic control of adult animals in vivo, are basically unknown due to the early embryonic lethality caused by disruption of *Tip60* gene[17].

In the current work, we show that mice expressing a *Tip60*[S86A] allele encoding the less active S86A-Tip60 mutant exhibit severe reduction in adiposity, and that postpartum *Tip60*[SA/SA] females fail to produce enough milk TAGs to maintain survival of newborn pups. In addition, we find that the translocalization of lipin 1 from cytosol to ER membranes, together with TAG synthesis, are intrinsically dependent on Tip60-mediated lipin 1 acetylation, which is suppressed by deacetylase Sirt1. These results highlight a fundamental role of Tip60 and lipin 1 acetylation in fat synthesis, and proper modulation of lipin 1 acetylation might be a promising strategy for the alleviation of obesity and its associated metabolic disorders/diseases.

## Results

***Tip60*[SA/SA] mice demonstrate lean phenotypes**. To explore the physiological functions of Tip60, we generated a knock-in mouse strain in which the wild-type (WT) *Tip60* allele is replaced by the *Tip60-S86A* mutant (serine-86 altered to alanine). The expressed Tip60[S86A] protein can no longer be phosphorylated and activated by glycogen synthase kinase-3 (GSK3), which leads to the attenuation of Tip60 aetyltransferase activity by almost 50%[15, 18] (Fig. 1a and Supplementary Fig. 1a–c). Mice homozygous for the *Tip60*[S86A] allele (*Tip60*[SA/SA] mice) were born at the expected Mendelian frequency (WT: *Tip60*[+/SA]: *Tip60*[SA/SA] = 137:280:131), and appeared normal at birth. However, compared to their WT or heterozygous littermates, *Tip60*[SA/SA] mice exhibited 20–30% reductions in body weight beginning at 3 weeks of age and throughout their lifetime under a normal chow diet (ND), independent of gender (Fig. 1b and Supplementary Fig. 1d). The difference in body weight became more pronounced when the mice were subjected to a high-fat diet (HFD) (60% fat) beginning at 6 weeks of age (Fig. 1b and Supplementary Fig. 1d). After 14 weeks on the HFD, the body weight of WT mice increased by more than 60%, but *Tip60*[SA/SA] mice maintained a similar body weight compared to the ND-fed mice (Fig. 1c), despite comparable food intake and fecal fat content among them (Supplementary Fig. 1e,f). The lower body weight in *Tip60*[SA/SA] mice was associated with smaller fat depots as well as reduced adipocyte size compared to WT (Fig. 1d–g and Supplementary Fig. 1g–i). Moreover, unlike WT mice, *Tip60*[SA/SA] mice on HFD did not develop hepatic steatosis (Fig. 1g, h, and Supplementary Fig. 1i), and maintained lower plasma TAG levels on both ND and HFD (Fig. 1i). Consistently, *Tip60*[SA/SA] mice were resistant to HFD-induced increases in plasma glucose and insulin (Fig. 1j, k), and exhibited greater glucose tolerance and insulin sensitivity as demonstrated by glucose tolerance test (GTT) and insulin tolerance test (ITT) (Fig. 1l–o). By analysis of covariance (ANCOVA), we found that the higher insulin sensitivity in *Tip60*[SA/SA] mice is likely due to lower body weight in these mice (Fig. 1p).

The reduced body weight in *Tip60*[SA/SA] mice was associated with increased energy expenditure (EE), lower respiratory exchange ratio (RER) (indicating a preference for fat utilization) and higher, albeit not statistically significant, physical activity compared to WT mice under short-term HFD (Fig. 1q and Supplementary Fig. 1j–l). Mice lacking enzymes of the TAG synthetic pathway, including Agpat2, lipin 1 and Dgat1, exhibit similar changes in EE and RER (Agpat2 or lipin 1 deficiency) and physical activity (Dgat1 deficiency)[8, 19, 20]. Skeletal muscles from the *Tip60*[SA/SA] mice also exhibited increased expression of the gene encoding carnitine palmitoyl transferase 1 β (Cpt1β), a rate-limiting enzyme for fatty acid oxidation (Supplementary Fig. 1m), consistent with an increased fatty acid oxidation in these mice. There was no evidence that *Tip60*[SA/SA] mice have altered expression of uncoupling protein 1 (Ucp1), a key player in thermogenesis in brown adipose tissue (Supplementary Fig. 1n). These results indicate that increased fat oxidation or impaired TAG synthesis may be responsible for the lean phenotypes of *Tip60*[SA/SA] mice in vivo.

**Essential role of Tip60 in fatty acid-induced TAG synthesis.** During the breeding of *Tip60*[SA/SA] mice, we found that more than 80% of both *Tip60*[SA/SA] and WT pups fostered by *Tip60*[SA/SA] females died within 3 days after birth. However, when they were cross-fostered by WT females, all pups survived (Fig. 2a). Lipids, primarily TAG, in the milk supply the majority of the calories required for neonatal growth in most mammals[21]. We found that

both the number of milk fat globules and TAG concentration were markedly reduced in the milk from *Tip60^SA/SA* females compared with those from WT females (Fig. 2b, c). In sections of mammary tissue, neutral lipid-staining droplets were rare in the apical regions of the epithelial cells and in the ductal lumens of

*Tip60^S86A* females, but were abundant in lactating WT females (Fig. 2d). Tip60 therefore appears to be crucial for the production of milk TAG.

To determine the molecular mechanisms by which Tip60 regulates TAG metabolism, we first knocked down *Tip60* in

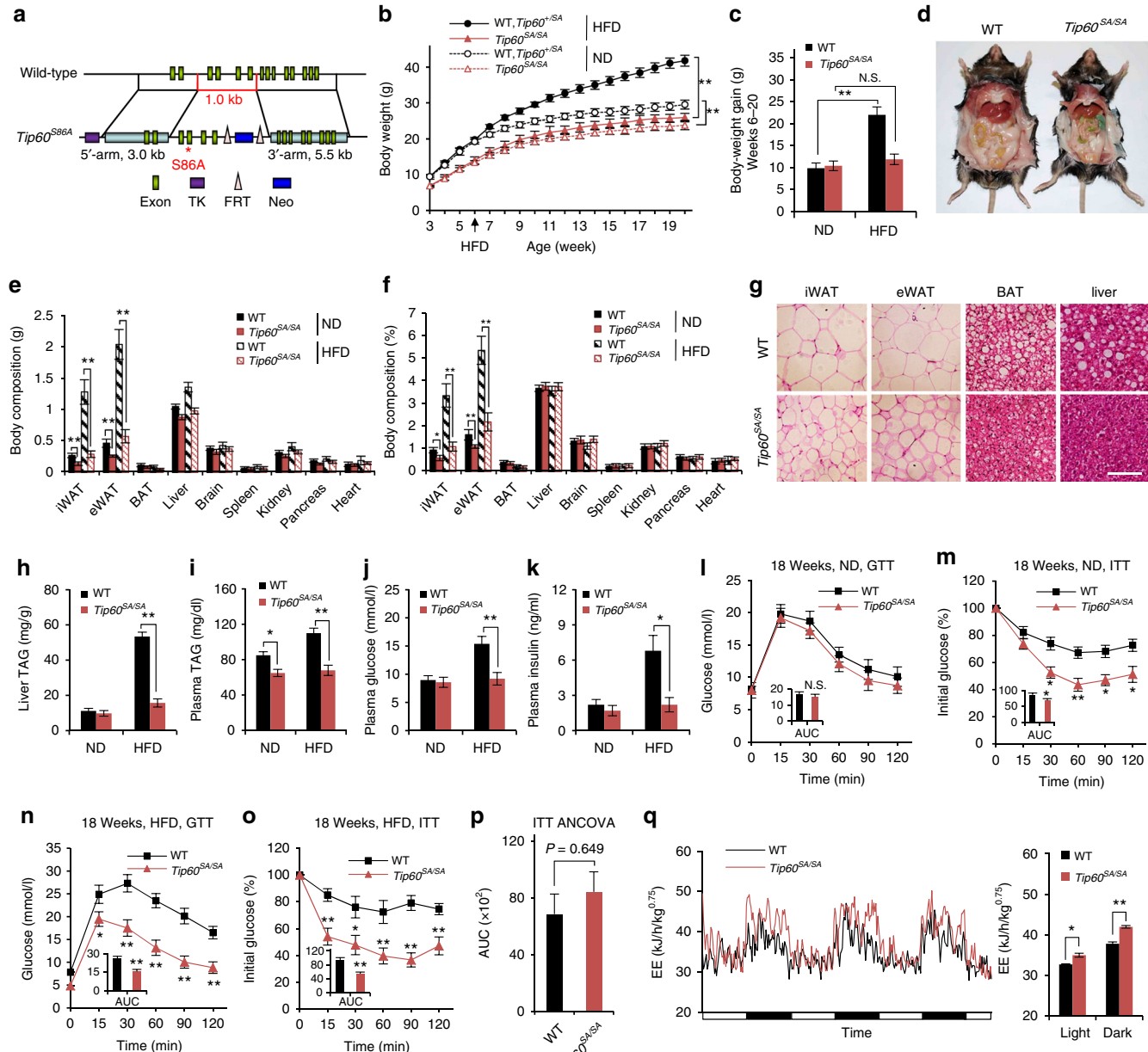

**Fig. 1** *Tip60^SA/SA* mice demonstrate lean phenotypes and are resistant to obesity-associated metabolic disorders. **a** Schematic diagram of the strategy for *Tip60^S86A* knock-in mice. **b** Body weights of wild-type (WT), *Tip60^+/SA* and *Tip60^SA/SA* mice fed with a normal chow diet (ND) (male, n = 20 per group) or high-fat diet (HFD, starting at 6 weeks of age) (male, n = 24 per group). **c** Weight gains of ND or HFD mice from **b**. **d** Representative photographs of WT and *Tip60^SA/SA* HFD mice at 18 weeks of age. **e** Weights of different tissues from 18-week-old WT and *Tip60^SA/SA* ND or HFD mice (male, n = 16 per group). iWAT inguinal white adipose tissue, eWAT epididymal WAT, BAT brown adipose tissue. **f** Percentages of the weights of different tissues as compared to their corresponding body weight. **g** Representative sections of iWAT, eWAT, BAT and liver from HFD 18-week-old male mice. Scale bar, 100 μm. **h–k** Levels of liver triacylglycerol (TAG) (**h**), plasma TAG (**i**), plasma glucose (**j**) and plasma insulin (**k**) in WT and *Tip60^SA/SA* ND (male, n = 8 per group) or HFD (male, n = 12 per group) mice fasted for 6 h. **l, m** Glucose tolerance test (GTT) (male, n = 12 per group) (**l**) and insulin tolerance test (ITT) (male, n = 10 per group) (**m**) in WT and *Tip60^SA/SA* ND mice. Inset graphs show area under curve (AUC). **n, o** GTT (male, n = 10 per group) (**n**) and ITT (male, n = 12 per group) (**o**) in WT and *Tip60^SA/SA* HFD mice. **p** The adjusted means of AUCs of **o** analyzed by ANCOVA, using body weight as the covariate. **q** Energy expenditure (EE) of WT and *Tip60^SA/SA* mice fed HFD for 1 week (male, n = 6 per group), before the mice diverge too much in body weight. For EE, data were collected for 3 consecutive days, expressed as adjusted means based on body weight to the power 0.75. EE is estimated with Weir equation (16.3 × VO$_2$ + 4.57 × VCO$_2$). Error bars denote SEM. Statistical analysis was performed by ANOVA followed by Tukey in **b**, **c**, **e**, **f**, **h–l** and **q** or by two-tailed unpaired Student's *t*-test in AUC graphs of **l–o**. *P < 0.05; **P < 0.01

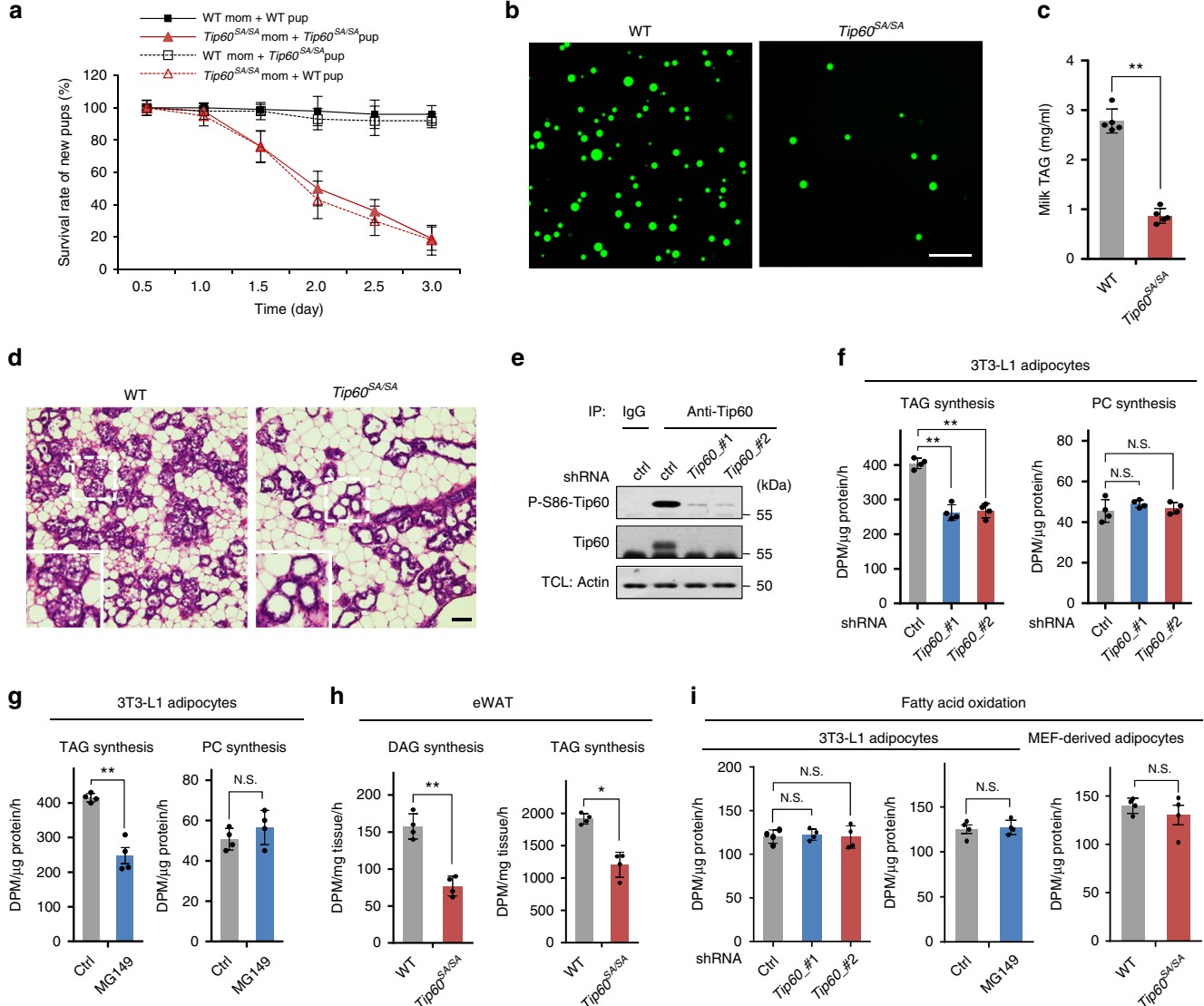

**Fig. 2** Tip60 plays an essential role in fatty acid-induced TAG synthesis. **a** Survival rates of $Tip60^{SA/SA}$ litters nursed by WT or $Tip60^{SA/SA}$ females ($n = 8$ litters for each group), and WT litters nursed by WT or $Tip60^{SA/SA}$ females ($n = 7$ litters for each group). **b** Representative BODIPY staining ($n = 3$ experiments) of milk fat globules (green) from the mammary glands of early lactating (day 2 postpartum) WT and $Tip60^{SA/SA}$ females. Scale bar, 50 μm. **c** Level of TAG in milk collected from WT and $Tip60^{SA/SA}$ females (day 2 postpartum, $n = 5$ per group). **d** Representative images of mammary glands from WT and $Tip60^{SA/SA}$ females at day 2 postpartum ($n = 3$ experiments), visualized by H&E staining. The WT mouse (left) has numerous fat globules in the apical regions of mammary epithelial cells and in the ductal lumens, but were rare in the $Tip60^{SA/SA}$ mouse (right). Scale bars, 50 μm. **e** Lentivirus-mediated knockdown of Tip60 in 3T3-L1 adipocytes. TCL total cell lysate. **f** TAG and phosphatidylcholine (PC) synthesis rates of 3T3-L1 adipocytes expressing control shRNA (ctrl) or shRNAs targeting Tip60 with ³H-labeled oleic acid (OA) treatment ($n = 4$ experiments). **g** TAG and PC synthesis rates of 3T3-L1 adipocytes treated with DMSO (ctrl) or Tip60 inhibitor MG149 ($n = 4$ experiments). **h** Rates of diacylglycerol (DAG) and TAG synthesis from ³H-OA in eWAT explants isolated from 6-week-old WT and $Tip60^{SA/SA}$ mice ($n = 4$ individuals per group). **i** Fatty acid oxidation rates of control (ctrl) or Tip60 knockdown or MG149-treated 3T3-L1 adipocytes or adipocytes derived from WT or $Tip60^{SA/SA}$ MEFs (day 8) ($n = 4$ experiments). Error bars denote SEM. Statistical analysis was performed by two-tailed unpaired Student's t-test in **c**, **g**, **h** and middle and right bar graph of **i**, by one-way ANONA followed by Tukey in **f** and left bar graph of **i**. *$P < 0.05$; **$P < 0.01$; N.S. not significant. Uncropped blots can be found in Supplementary Fig. 6

differentiated 3T3-L1 adipocytes by two separate short hairpin RNAs (shRNAs) (Fig. 2e). We found that depletion of Tip60 strongly impeded the synthesis of TAG as determined by using ³H-labeled oleic acid (OA), while phosphatidylcholine (PC) synthesis was unaffected (Fig. 2f). Similar results were obtained after treating 3T3-L1 adipocytes with MG149[22], a small-molecule inhibitor of Tip60 (Fig. 2g). Moreover, the incorporation of ³H-labeled OA into DAG or TAG was markedly reduced in epididymal white adipose tissue (eWAT) explants from $Tip60^{SA/SA}$ mice compared to WT controls (Fig. 2h and Supplementary Fig. 2a). We also checked the ability of primary

mouse embryonic fibroblasts (MEFs) from $Tip60^{SA/SA}$ and WT mice to differentiate into adipocytes. Surprisingly, primary MEFs or adipose stromal vascular fibroblasts (SVFs) from $Tip60^{SA/SA}$ mice demonstrated a similar capacity in adipogenic differentiation compared to WT MEFs, with robust induction of adipogenic transcription factors PPARγ and CCAAT/enhancer binding protein β (C/EBPβ), and key adipocyte proteins including fatty acid synthase (FASN), stearoyl-CoA desaturase-1 (SCD1) and perilipin in both WT and $Tip60^{SA/SA}$ MEFs/SVFs (Supplementary Fig. 2b–d). However, adipocytes derived from the $Tip60^{SA/SA}$ MEFs exhibited compromised accumulation of labeled DAG and

TAG after [3]H-OA loading in comparison to WT MEFs (Supplementary Fig. 2e). Meanwhile, no difference in the rate of fatty acid oxidation was detected in cells with *Tip60* knockdown, MG149 treatment or *Tip60*[S86A] knock-in compared to that in control cells (Fig. 2i). These results indicate that the decreased TAG content caused by Tip60 depletion or inhibition most likely results from a block of TAG synthesis rather than a decrease of adipogenesis or an increase of lipid consumption, although it is possible that other aspects of Tip60 function may influence adipogenesis[23, 24].

**Tip60 directly interacting with and acetylating lipin 1.** We next explored whether Tip60 may influence TAG synthesis through regulating any of the TAG biosynthetic enzymes (Fig. 3a). It was found that exogenous Myc-tagged Tip60 strongly and specifically interacted with Flag-tagged lipin 1 (Fig. 3b), which catalyzes the dephosphorylation of PA to DAG, the penultimate step in TAG

biosynthesis[8, 9]. We then asked whether lipin 1 is an acetylation substrate of Tip60. In line with the interaction results, Tip60 acetylated lipin 1 but not other TAG biosynthetic enzymes such as AGPAT1 or DGAT1 in vitro (Fig. 3c and Supplementary Fig. 3a). Importantly, the amount of acetylated lipin 1 was significantly increased upon OA treatment in 3T3-L1 adipocytes, which was almost entirely abolished by shRNA-mediated knockdown of *Tip60* (Fig. 3d). Treatment of cells with Tip60 inhibitor MG149 also reduced lipin 1 acetylation (Fig. 3e). In addition, S86A-Tip60 mutant showed much compromised ability in acetylating lipin 1 compared to WT-Tip60 (Fig. 3f), and eWAT from *Tip60*[SA/SA] mice exhibited attenuated lipin 1 acetylation in comparison with that from WT mice under both ND and HFD (Fig. 3g), which is consistent with Tip60-Ser[86] phosphorylation being necessary for its efficient acetyltransferase activity[15, 18, 25]. Interestingly, we found that, compared to WT-Tip60, S86A-Tip60 mutant also demonstrated decreased binding affinity for

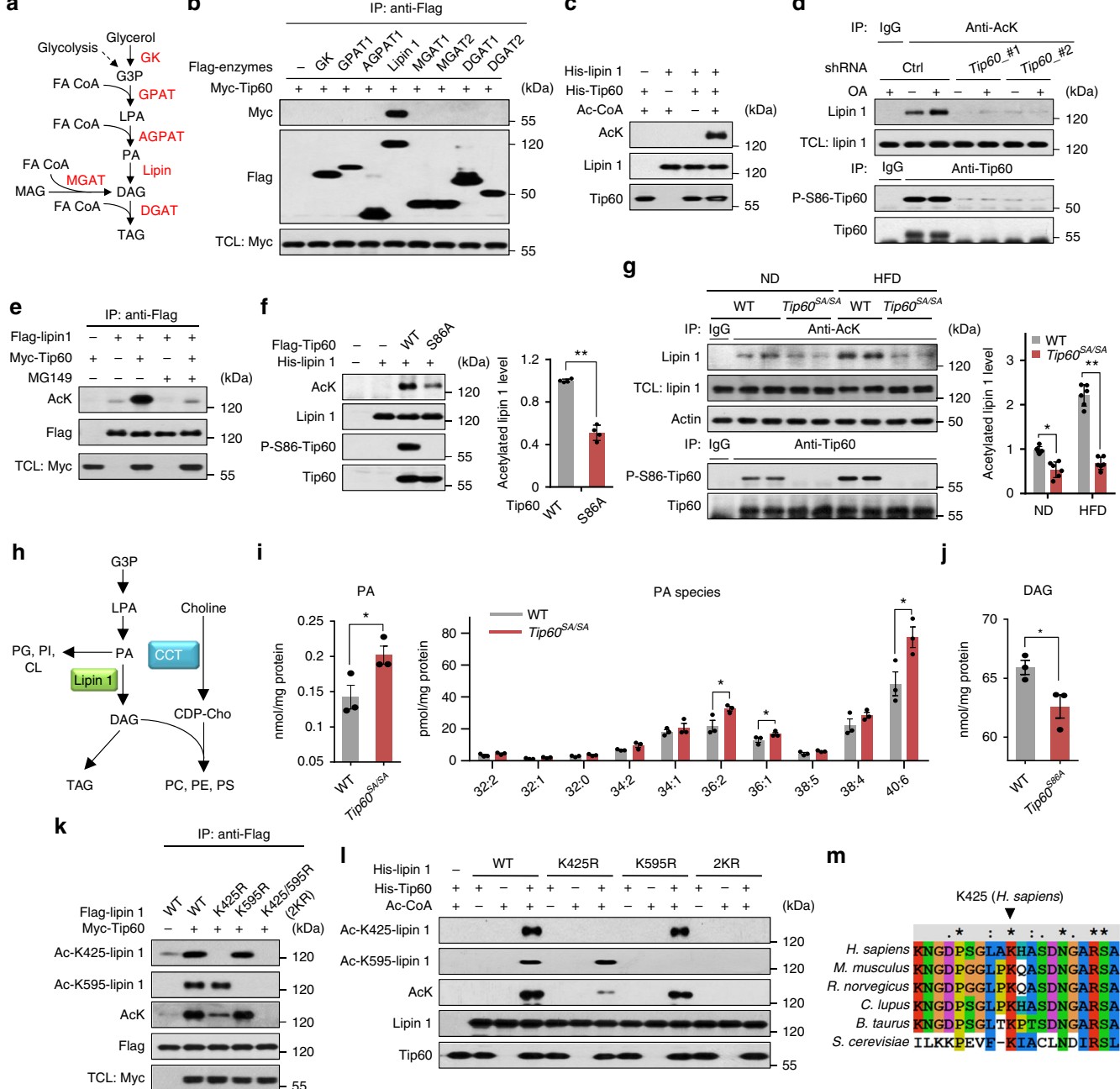

lipin 1 (Supplementary Fig. 3b), which may further contribute to its diminished ability in acetylating lipin 1.

To further confirm a functional relationship between Tip60 and lipin 1, an untargeted lipidomic profiling of nearly all kinds of lipid species in eWAT from WT and $Tip60^{SA/SA}$ mice by liquid chromatography-mass spectrometry (LC-MASS) was conducted. The results showed that the total level of PA, substrate of lipin 1, was increased by ~40%, while the level of DAG, product generated by lipin 1, was conversely decreased in the eWAT of $Tip60^{SA/SA}$ mice compared with those from WT mice (Fig. 3h–j). In addition, phosphatidylglycerol (PG) and phosphatidylinositol (PI) which directly come from PA were among the most elevated lipid species in eWAT of $Tip60^{SA/SA}$ (Supplementary Fig. 3c). Meanwhile, other phospholipids, such as PC and PE, demonstrated fewer alternations between two genotypes (Supplementary Table 1). These results support a critical role of Tip60 activity in the catalytic function of lipin 1 in vivo.

We then sought to identify the amino acid residues that are acetylated by Tip60. In vitro acetylated lipin 1 was subjected to mass spectrometry and nine lysine residues were found to be candidate acetylation sites targeted by Tip60 (Supplementary Fig. 3d,e). To validate these sites, we created mutants harboring arginine residues in place of these lysine residues, either singly or in combination. We found that combined mutation of $Lys^{425}$ and $Lys^{595}$ (2KR) on lipin 1 completely abolished its acetylation by Tip60 both in vivo and in vitro (Fig. 3k, l). Moreover, Tip60-mediated acetylation of lipin 1 on $Lys^{425}$ and $Lys^{595}$ was confirmed by antibodies that specifically recognize the corresponding acetylated lysines (Fig. 3k, l). Sequence alignment indicates that both $Lys^{425}$ and $Lys^{595}$ in lipin 1 are conserved in higher eukaryotes, and the $Lys^{425}$ site is even conserved back to Saccharomyces cerevisiae (Fig. 3m and Supplementary Fig. 3f).

**Sirt1 deacetylates lipin 1 and inhibits the synthesis of TAG.** We next searched for the deacetylase that is responsible for lipin 1 deacetylation. Treatment of HEK293T cells expressing exogenous lipin 1 with trichostatin A (TSA), a class I and II histone deacetylase (HDAC) inhibitor, did not affect lipin 1 acetylation. In contrast, nicotinamide (NAM), a class III HDAC (sirtuins) inhibitor, strongly increased levels of acetylated lipin 1 (Fig. 4a), suggesting that certain sirtuins may act to deacetylate lipin 1. By co-transfecting lipin 1 and Tip60 with each of the seven sirtuins

separately into HEK293T cells, we found that expression of Sirt1 specifically reduced the acetylation of lipin 1 to background levels (Fig. 4b). An in vitro deacetylation assay confirmed that lipin 1 was deacetylated in an $NAD^+$-dependent manner by WT-Sirt1, but not the catalytically inactive H363Y-Sirt1 mutant (Fig. 4c). In addition, inhibition of Sirt1 by shRNA-mediated knockdown or EX527, a specific Sirt1 inhibitor[26], remarkably increased the amount of acetylated lipin 1 accompanied by increased rates of DAG and TAG synthesis in 3T3-L1 adipocytes (Fig. 4d–g). To clarify whether suppression of TAG synthesis by Sirt1 is mediated by lipin 1, we treated the 3T3-L1 adipocytes expressing either control shRNA or shRNA targeting $Lpin1$ with Sirt1 inhibitor EX527. It was found that inhibition of Sirt1 could no longer increase TAG/DAG synthesis in cells expressing shRNA targeting $Lpin1$ (Fig. 4h), indicating that the suppression of TAG synthesis by Sirt1 is indeed dependent on lipin 1. Together, these data provide strong evidence that Sirt1 deacetylates lipin 1 and negatively regulates fat synthesis.

Next, we knocked down $Lpin1$ in either WT or $Tip60^{SA/SA}$ adipocytes, and analyzed the rates of TAG/DAG synthesis. It was found that knockdown of $Lpin1$ strongly diminished the differences of TAG/DAG synthesis between WT and $Tip60^{SA/SA}$ adipocytes (Fig. 4i), indicating that Tip60 regulates TAG/DAG synthesis through lipin 1.

**Acetylation of lipin 1 facilitates its translocation to ER.** It is known that, in response to fatty acid such as OA, lipin 1 does not exhibit altered PAP activity, but rather becomes dephosphorylated and translocates from cytoplasm to ER-associated (microsomal) membranes where DAG/TAG are generated[9]. We found that, similar to phosphorylation[9, 11], acetylation did not affect lipin 1 PAP activity such assayed (Supplementary Fig. 4a). We thus examined if Tip60-catalyzed acetylation of lipin 1 regulated its translocation to ER. It was found that knockdown of Tip60 or treatment of 3T3-L1 adipocytes with MG149 strongly diminished the levels of microsomal lipin 1 under both OA-treated and -untreated conditions (Fig. 5a, b, and Supplementary Fig. 4b). Similarly, lower levels of microsomal lipin 1 were detected in adipocytes derived from $Tip60^{SA/SA}$ MEFs compared to those from WT MEFs (Fig. 5c). In contrast, inhibition of Sirt1 by shRNA or EX527 in 3T3-L1 adipocytes led to increased lipin 1 ER localization (Fig. 5d and Supplementary Fig. 4c). Moreover, compared to WT-lipin 1, the unacetylatable 2KR-lipin 1 exhibited

**Fig. 3** Tip60 regulates lipid biosynthesis by directly interacting with and acetylating lipin 1. **a** The TAG biosynthesis pathway in mammalian cells. **b** Association of Flag-tagged enzymes in TAG synthesis pathway with Myc-tagged Tip60. Protein extracts of HEK293T after transfection were immunoprecipitated with antibody to Flag and immunoblotted as indicated. **c** Lipin 1 was acetylated by Tip60 in vitro. Bacterially expressed His-lipin 1 was incubated with His-Tip60 in the presence or absence of acetyl-CoA (Ac-CoA) and immunoblotted as indicated. **d** Tip60-dependent acetylation of lipin 1. 3T3-L1 adipocytes expressing control shRNA (ctrl) or shRNAs targeting $Tip60$ were treated with or without OA for 3 h. Acetylated proteins were immunoprecipitated with antibody to acetylated lysine, followed by immunoblotting. **e** Tip60 inhibitor MG149 abrogates the acetylation of lipin1 mediated by Tip60. Flag-tagged lipin 1 co-transfected with or without Tip60 in HEK293T cells treated or untreated with MG149 for 3 h were immunoprecipitated by antibody to Flag and immunoblotted as indicated. **f** Reduced acetyltransferase activity of Tip60-S86A towards lipin 1. Flag-tagged Tip60 or its S86A mutant expressed in HEK293 cells was immunoprecipitated with antibody to Flag. In vitro acetylation assays were performed using purified His-lipin 1 as substrate ($n = 4$ experiments). **g** Tip60-dependent acetylation of lipin 1 in vivo. eWAT from WT or $Tip60^{SA/SA}$ ND or HFD mice was immunoprecipitated by using antibody to acetylated lysine or Tip60, followed by immunoblotting. Each lane represents a different individual ($n = 3$ experiments). **h** Biosynthesis of phospholipids and neutral lipids from glycerol-3-phosphate (G3P) in mammalian cells. **i** Total levels of PA (left) and individual PA species (right) normalized to cellular protein in eWAT from 6-week-old WT and $Tip60^{SA/SA}$ mice. The molecular species of PA are indicated as total number of carbons/number of double bonds ($n = 3$ individuals for each group). **j** Total levels of DAG normalized to cellular protein in eWAT from 6-week-old WT and $Tip60^{SA/SA}$ mice ($n = 3$ individuals for each group). **k, l** Identification of acetylation sites on lipin 1 targeted by Tip60. WT-lipin 1 or its lysine-to-arginine (KR) mutants were co-expressed with or without Tip60 in HEK293T cells, immunoprecipitated by antibody to Flag and immunoblotted as indicated (**k**). His-tagged WT-lipin 1 or its KR mutants were incubated with Tip60 in the presence or absence of Ac-CoA and immunoblotted as indicated (**l**). **m** Sequence alignment of the residues flanking $Lys^{425}$ across different species. Arrowheads point to the $Lys^{425}$ residue corresponding to human lipin 1. Error bars denote SEM. Statistical analysis was performed by ANOVA followed by Tukey in **g** or by two-tailed unpaired Student's $t$-test in **f**, **i** and **j**. *$P < 0.05$; ** $P < 0.01$. Uncropped blots can be found in Supplementary Fig. 6

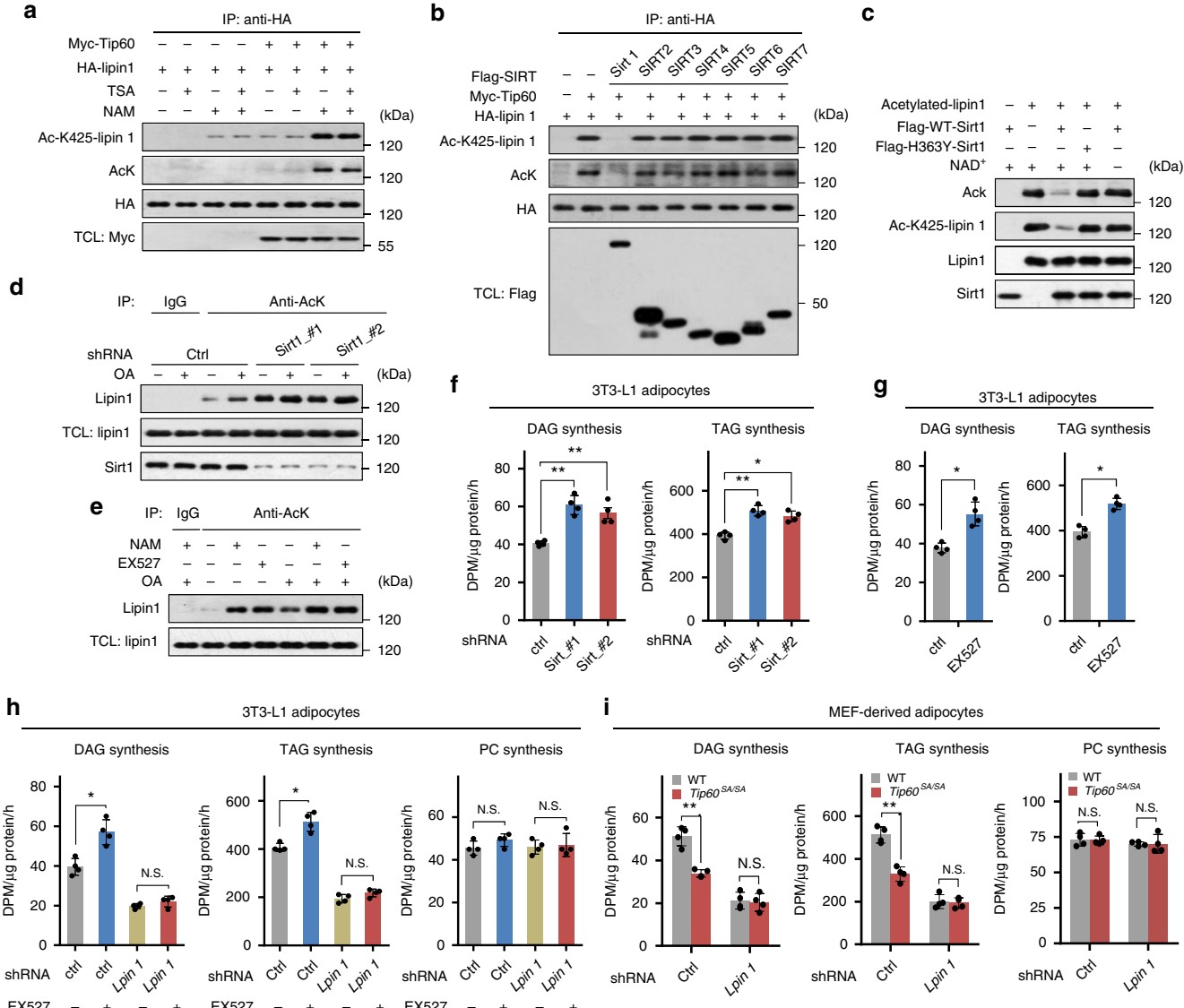

**Fig. 4** Sirt1 deacetylates lipin 1 and represses the synthesis of DAG and TAG. **a** Sirtuins inhibitor nicotinamide (NAM) increases the amount of acetylated lipin 1. Lysates of HEK293T cells transfected with HA-lipin 1 and/or Tip60 treated with trichostatin A (TSA) (a class I and II HDAC inhibitor) or NAM for 6 h were immunoprecipitated with antibody to HA, followed by immunoblotting. **b** Deacetylation of lipin 1 by co-expression with Sirtuins. Lysates of HEK293T cells transfected with HA-lipin 1, Tip60 and Flag-tagged Sirt1–7 as indicated were immunoprecipitated with antibody to HA, followed by immunoblotting. **c** In vitro deacetylation of lipin 1 by Sirt1. Acetylated-lipin1 were incubated with Flag-tagged WT-Sirt1 or the inactive H363Y mutant (expressed and immunoprecipitated from HEK293T cells) in the presence or absence of NAD$^+$ for 1 h, followed by immunoblotting. **d** Knockdown of *Sirt1* increases the amount of acetylated lipin 1. 3T3-L1 adipocytes expressing control shRNA (ctrl) or shRNAs targeting *Sirt1* were treated with or without OA for 3 h. Acetylated proteins were immunoprecipitated with antibody to acetylated lysine. **e** Increased lipin 1 acetylation by Sirt1 inhibitor EX527. 3T3-L1 adipocytes were treated with or without NAM, EX527 or OA for 3 h and analyzed as in **d**. **f, g** Knockdown of *Sirt1* by shRNAs (**f**) ($n = 4$ experiments) or inhibition of Sirt1 activity by EX527 (**g**) ($n = 4$ experiments) promotes DAG and TAG synthesis in 3T3-L1 adipocytes. **h** TAG, DAG and PC synthesis rates of 3T3-L1 adipocytes expressing control shRNA (ctrl) or shRNA targeting *Lpin1* with or without EX527 treatment ($n = 4$ experiments). **i** TAG/DAG synthesis rates of WT or *Tip60*$^{SA/SA}$ MEF-derived adipocytes expressing control shRNA (ctrl) or shRNA targeting *Lpin1*. Error bars denote SEM. Statistical analysis was performed by ANOVA followed by Tukey in **f, h, i** or by two-tailed unpaired Student's *t*-test in **g**. *$P < 0.05$, **$P < 0.01$. Uncropped blots can be found in Supplementary Fig. 6

much lower sensitivity in the translocation process after OA treatment in *Lpin1* knockdown adipocytes (Fig. 5e, f). In line with the acetylation-dependent lipin 1 ER translocation, the DAG and TAG synthesis rates were markedly reduced in *Lpin1* knockdown adipocytes expressing 2KR-lipin 1 compared to those in cells expressing WT-lipin 1 (Fig. 5g). Collectively, our data indicate that Tip60-dependent lipin 1 acetylation is critical for the translocation of lipin 1 to ER membranes as well as for DAG generation and TAG synthesis.

We noticed that there is still some amount of basal lipin 1 located on the ER membrane in cells with *Tip60* knockdown, inhibition or *Tip60*$^{SA}$ knock-in (Fig. 5b, c and Supplementary Fig. 4b), suggesting that there may exist other mechanisms for lipin ER localization. In agreement with another report[27], we found that lipin 1 could directly bind to artificial liposomes that consist of only PC and PA in vitro in an acetylation-independent manner (Supplementary Fig. 4d), which could contribute to the basal localization of lipin1 on ER.

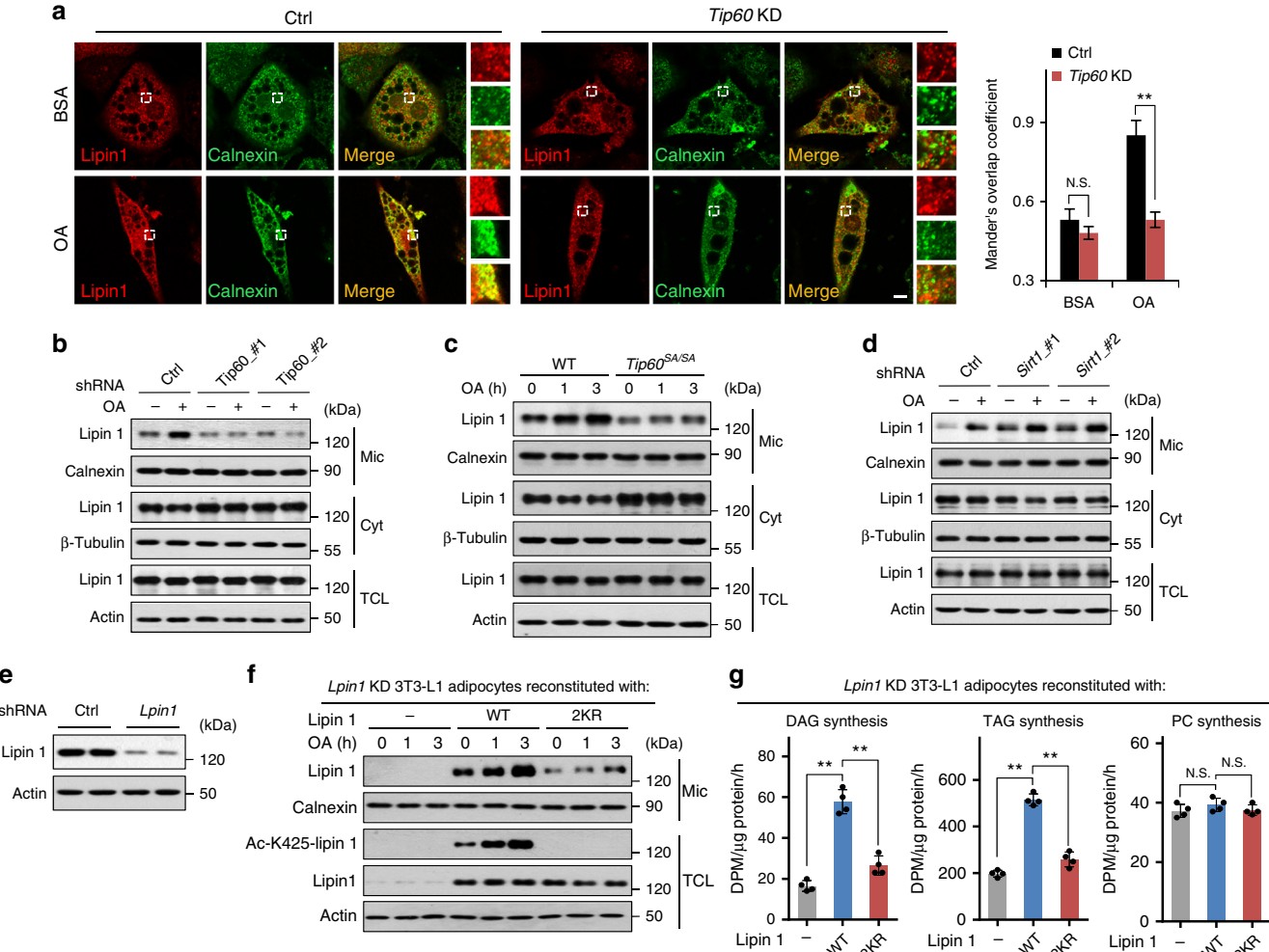

**Fig. 5** Tip60-mediated lipin 1 acetylation leads to lipin 1 translocation from cytosol to ER membranes. **a** Attenuated translocation of lipin 1 from cytosol to ER membranes after Tip60 depletion. Representative images ($n = 3$ replicate experiments) of 3T3-L1 adipocytes expressing control shRNA (ctrl) or shRNA targeting *Tip60* (*Tip60* KD) treated with BSA or OA for 2 h. Calnexin is an ER/microsomal marker. Mander's overlap coefficients between lipin 1 and Calnexin were graphed ($n = 30$ for each group). Scale bar, 10 μm. **b** Depletion of Tip60 impairs lipin 1 ER localization. Western blotting of proteins in subcellular fractions of 3T3-L1 adipocytes expressing control shRNA (ctrl) or shRNAs targeting *Tip60* treated with or without OA for 3 h. Calnexin, microsomal (Mic) marker; β-tubulin, cytosol (Cyt) marker. **c** Impairment of lipin 1 ER localization in adipocytes derived from *Tip60*$^{SA/SA}$ MEFs. Adipocytes (differentiation day 8) derived from WT or *Tip60*$^{SA/SA}$ MEFs were treated with OA for 1–3 h and analyzed by fractionation, followed by western blotting. **d** Depletion of Sirt1 promotes lipin 1 ER localization. **e** Knockdown of *Lpin1* in 3T3-L1 adipocytes. Western blotting of proteins of 3T3-L1 adipocytes expressing control shRNA (ctrl) or shRNA targeting *Lpin1*. **f, g** Acetylation-dependent ER localization of lipin 1 and DAG/TAG synthesis. Western blotting of proteins (**f**) or DAG, TAG and PC synthesis rates (**g**) of 3T3-L1 adipocytes expressing shRNA targeting *Lpin1* reconstituted with vector, WT-lipin 1 or 2KR-lipin 1 after OA treatment ($n = 4$ experiments). Error bars denote SEM. Statistical analysis was performed by ANOVA followed by Tukey in **a**, **g**. \*\**P* < 0.01; N.S. not significant. Uncropped blots can be found in Supplementary Fig. 6

**Reciprocal association of lipin 1 with Sirt1 and Tip60**. Since dephosphorylation of lipin 1 during fatty acid treatment also promotes its ER translocation[9], we investigated possible interplay between dephosphorylation and acetylation of lipin 1 by examining the phosphorylation of 2KR-lipin 1 and the acetylation of the unphosphorylatable lipin 1 mutant[12]. We found that lipin 1 dephosphorylation strongly boosted its acetylation by Tip60, whereas the acetylation process did not turn around to affect the dephosphorylation process (Fig. 6a, b). In addition, Tip60 exhibited increased affinity for the unphosphorylatable lipin 1 mutant compared to WT-lipin 1 (Fig. 6a). Consistently, co-expression of lipin 1 phosphatase complex, C-terminal domain nuclear envelope phosphatase 1 (CTDNEP1) and nuclear envelope phosphatase 1-regulatory subunit 1 (NEP1-R1)[28], in combination with Tip60 and lipin 1 in HEK293T cells, led to lipin 1 dephosphorylation and promoted its acetylation by Tip60

(Supplementary Fig. 4e). By contrast, lipin 1 dephosphorylation reduced its association with Sirt1 (Fig. 6c). Moreover, Tip60 and Sirt1 competed with one another for binding to lipin 1 (Fig. 6d). In line with these results, OA treatment promoted lipin 1 interaction with Tip60 and attenuated its association with Sirt1 in 3T3-L1 adipocytes (Fig. 6e), leading to higher levels of acetylated lipin 1 (Fig. 3d and Fig. 4d). Together, these results indicate that Tip60 and Sirt1 associate with lipin 1 in a reciprocal manner by sensing lipin 1 phosphorylation levels, so as to dynamically regulate the acetylation/deacetylation of lipin 1.

Finally, to determine the contribution of lipin 1 acetylation in Tip60-mediated regulation of lipid synthesis, we examined the TAG/DAG production rate in *Tip60*$^{SA/SA}$ adipocytes expressing WT-lipin 1, 2KR or constitutive acetylated mimetic (2KQ) mutant of lipin 1. It was found that while *Tip60*$^{SA/SA}$ adipocytes expressing WT-lipin 1 demonstrated decreased TAG/DAG

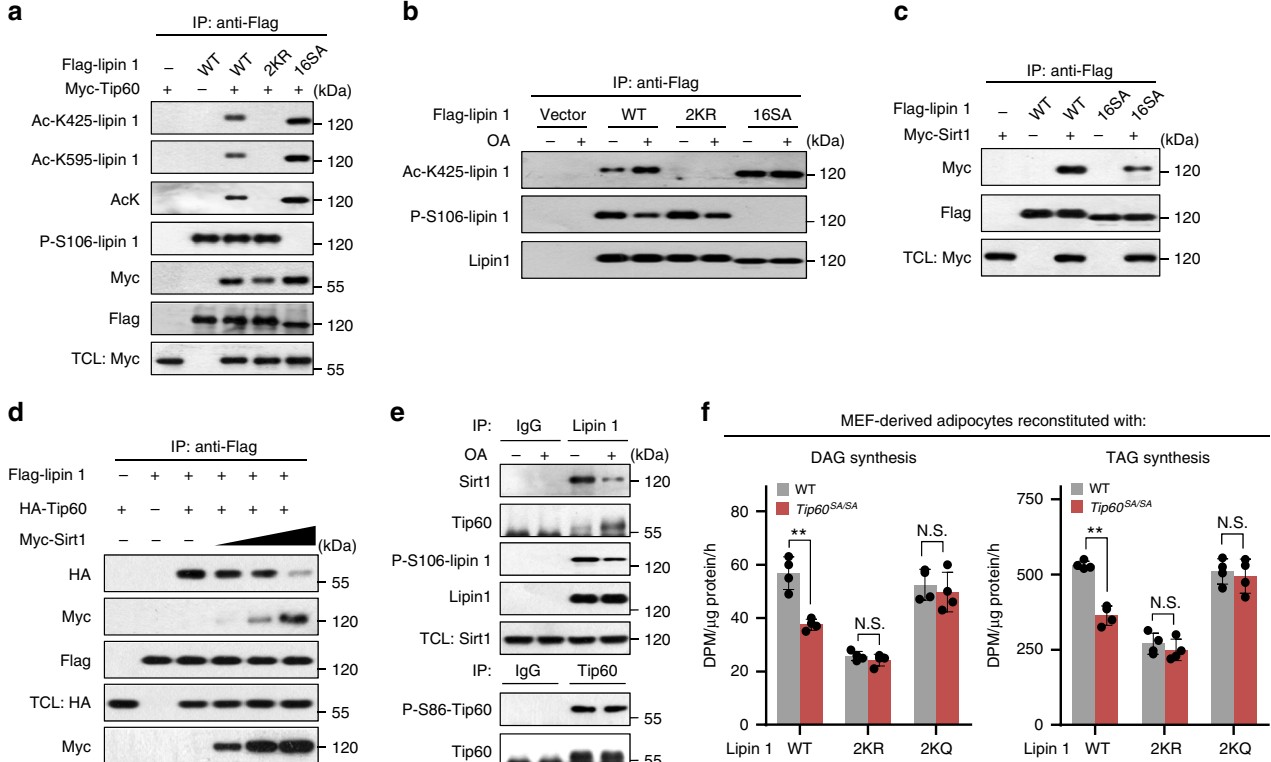

**Fig. 6** Lipin 1 acetylation senses fatty acid abundance by its reciprocal association with Sirt1 and Tip60. **a** Lipin 1 dephosphorylation promotes its acetylation by Tip60. Flag-tagged WT-lipin 1 or 2KR-lipin1 or 16SA-lipin 1 was co-expressed with or without Tip60 in HEK293 cells, immunoprecipitated by antibody to Flag, followed by immunoblotting. **b** Dephosphorylation of lipin 1 increases its acetylation by Tip60. HEK293 cells transfected with vector, WT-lipin 1 or 2KR-lipin 1 or 16SA-lipin 1 (phosphorylation-defective) were treated with or without OA for 3 h, immunoprecipitated with antibody to Flag and immunoblotted as indicated. **c** Dephosphorylation of lipin 1 attenuates its association with Sirt1. HEK293T cells were transfected with Flag-lipin 1 and Myc-Sirt1, immunoprecipitated with antibody to Flag and immunoblotted as indicated. **d** Tip60 and Sirt1 compete with each other in the interaction with lipin 1. HEK293T cells were transfected with Flag-lipin 1, HA-Tip60 and varied amounts of Myc-Sirt1, immunoprecipitated with antibody to Flag and immunoblotted as indicated. **e** Interaction of endogenous Tip60 and Sirt1 with lipin 1. Lysates of 3T3-L1 adipocytes treated with or without OA for 3 h were immunoprecipitated by antibody to lipin 1 or Tip60 and analyzed by immunoblotting. **f** TAG/DAG production rates of WT or $Tip60^{SA/SA}$ adipocytes expressing shRNA targeting Lpin1 with reintroduction of WT-lipin 1, acetylation-defective (2KR) or acetylated mimetic (2KQ) mutant of lipin 1 (n = 4 experiments). Error bars denote SEM. Statistical analysis was performed by ANOVA followed by Tukey in **f**. *P < 0.05; N.S. not significant. Uncropped blots can be found in Supplementary Fig. 6

synthesis rate compared to that in WT adipocytes, similar TAG/DAG synthesis rates were detected in $Tip60^{SA/SA}$ adipocytes expressing 2KR or 2KQ forms of lipin 1 compared to those in the corresponding WT adipocytes (Fig. 6f), indicating that Tip60-mediated lipin 1 acetylation plays a major role in determining the differences of TAG/DAG synthesis between WT and $Tip60^{SA/SA}$ adipocytes.

**Essential role of Tip60 in TAG synthesis in S. cerevisiae.** As both LPIN1 and Tip60 are highly conserved genes in eukaryotes[9, 13], we asked if the mechanism of Tip60-dependent lipin 1 acetylation in TAG synthesis is evolutionarily conserved in distantly related organisms such as S. cerevisiae. By analyzing loss-of-function mutants of catalytic subunits of all eight acetyltransferases in S. cerevisiae, we identified ESA1, the yeast ortholog of Tip60, as the only acetyltransferase that is required for TAG accumulation (Fig. 7a–c and Supplementary Fig. 5a,b). Co-expression of ESA1 with each of the TAG biosynthetic enzymes from S. cerevisiae in HEK293T cells specifically induced the acetylation of PAH1, the yeast ortholog of lipin 1, but not other enzymes tested (Fig. 7d, e). PAH1 plays an essential role in TAG synthesis throughout growth in S. cerevisiae[29, 30]. Two acetylation sites, Lys[496] and Lys[801], were found as the major acetylation sites on PAH1 targeted by ESA1 (Fig. 7f and Supplementary Fig. 5c).

Importantly, PAH1 knockout (pah1Δ) strains reconstituted with the acetylation-defective mutant (K496/801R) demonstrated severely decreased TAG accumulation compared to that in strains reconstituted with WT-PAH1 (Fig. 7g–i and Supplementary Fig. 5d). It is also noteworthy that the Lys[496] site on PAH1 is exactly the conserved acetylation site for Lys[425] on lipin 1 (Fig. 3m). These results reveal that PAH1 acetylation, most likely via the Tip60 ortholog ESA1, plays a conserved role in TAG storage in S. cerevisiae.

## Discussion

It is becoming evident that multiple cellular metabolic processes, such as glucose metabolism and autophagy, are intimately controlled by protein acetylation[31–33]. However, how the acetylation of diverse proteins, especially key metabolism modulators, connects extracellular nutrient signals to biological outcomes remains ill-defined. In this study, by generating and analyzing a severely lean mouse model that harbors homozygous knock-in of $Tip60^{S86A}$, and exploring the underlying molecular mechanisms in multiple cellular and organismal systems, we have discovered a signaling pathway that links fatty acid sensing to DAG generation and TAG synthesis. In this pathway, the interaction between Sirt1 and lipin 1 is attenuated after lipin 1 dephosphorylation in response to fatty acid stimulation. On the other hand,

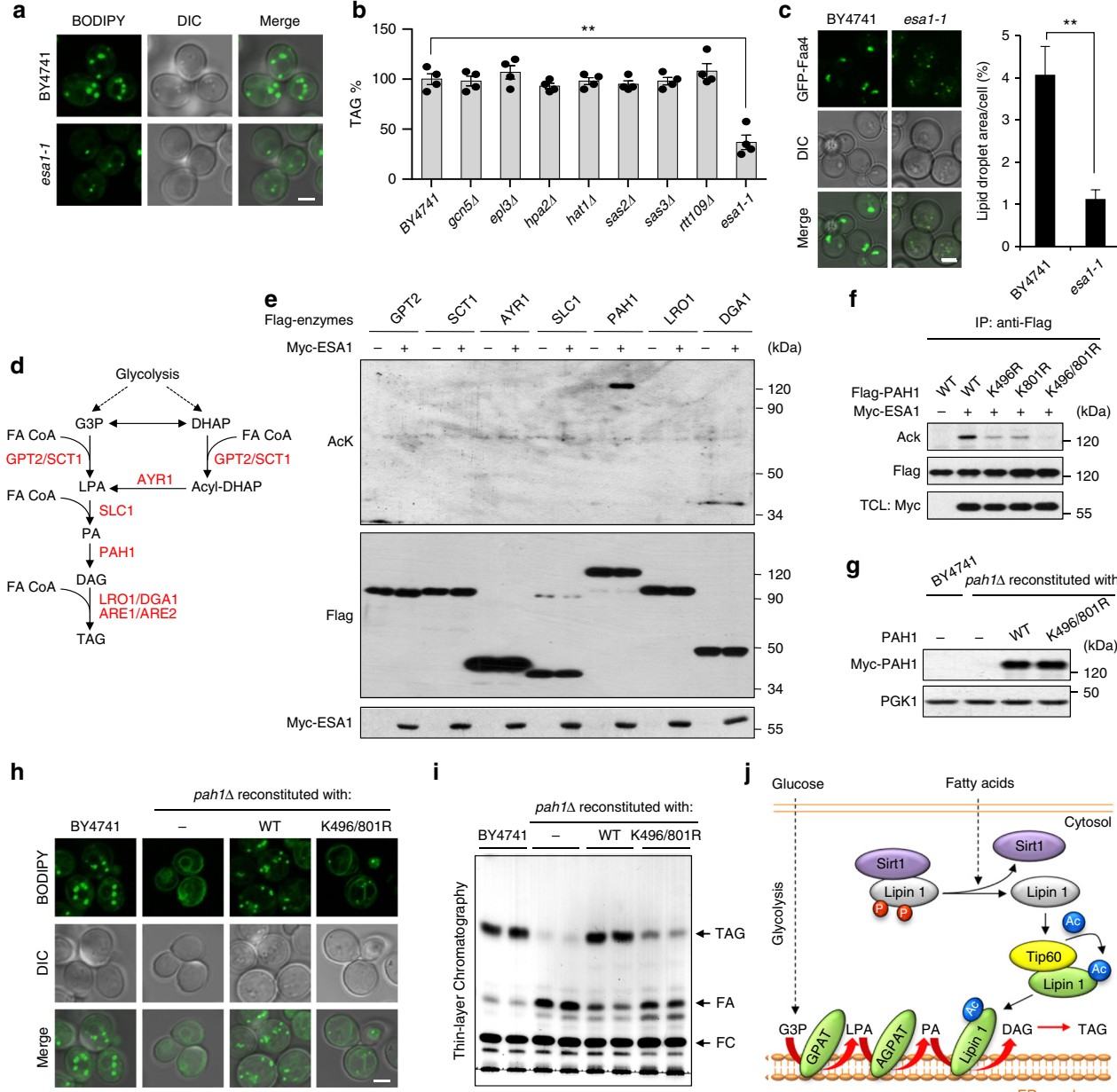

**Fig. 7** Essential roles of Tip60 and Tip60-mediated lipin 1 acetylation in TAG synthesis in *S. cerevisiae*. **a** Representative images (*n* = 3 replicate experiments) of BODIPY staining of WT (BY4741) and *ESA1* (yeast Tip60) loss-of-function mutant (*esa1-1*, an auxin-sensitive mutant strain of *ESA1*) cells grown in synthetic complete (SC) medium. DIC differential interference contrast microscopy images. Scale bar, 2 μm. **b** Identification of *ESA1* as essential gene for TAG accumulation in yeast. TAG was extracted from cells grown in SC medium, separated by thin-layer chromatography (TLC) and quantified (*n* = 4 experiments). **c** Decreased lipid droplet (LD) formation in *esa1-1* mutant strain. Representative images (*n* = 3 replicate experiments) of cells expressing chromosomally GFP-tagged fatty acid activation protein 4 (Faa4) (a LD marker protein) were grown in SC medium and analyzed (*n* = 100 cells quantified for each group). Scale bar, 2 μm. **d** TAG synthesis pathway in *S. cerevisiae*. **e** ESA1 specifically acetylates PAH1 but not other enzymes in the TAG synthesis pathway. Cell lysates of HEK293T cells transfected as indicated were immunoprecipitated with antibody to Flag, followed by immunoblotting. **f** Identification of acetylation sites on PAH1 (yeast lipin 1) targeted by ESA1. Flag-tagged WT-PAH1 or its KR mutants were co-expressed with or without ESA1 in HEK293T cells, immunoprecipitated by antibody to Flag and immunoblotted as indicated. **g** Protein expression levels of *pah1Δ* cells reconstituted with Myc-tagged WT-PAH1 or K496/801R-PAH1. Equal amounts of cell lysates from yeast cells as indicated were analyzed by immunoblotting. **h** Representative images (*n* = 3 replicate experiments) of BODIPY staining of WT (BY4741) and *PAH1* knockout (*pah1Δ*) cells reconstituted with vector, WT-PAH1 or K496/801R-PAH1. Scale bar, 2 μm. **i** Representative TLC result showing the TAG levels of WT or *PAH1* knockout (*pah1Δ*) cells reconstituted with vector, WT-PAH1 or K496/801R-PAH1. **j** Model of Tip60/Sirt1-regulated lipin 1 acetylation and TAG biosynthesis. Error bars denote SEM. Statistical analysis was performed by ANOVA followed by Tukey in **b** or by two-tailed unpaired Student's *t*-test in **c**. **\*\****P* < 0.01. Uncropped blots can be found in Supplementary Fig. 6

acetyltransferase Tip60 interacts with and directly acetylates lipin 1, promoting its translocation from cytoplasm to ER membranes. Upon translocation, lipin 1 positions itself into the TAG biosynthesis chain[34] and catalyzes the conversion of PA to DAG, feeding into the synthesis of TAG (Fig. 7g). In line with a positive role for lipin 1 acetylation in fat synthesis is the previous observation that mice with adipose tissue-specific knockout of *Sirt1*, the main deacetylases for lipin 1, are more prone to become obese on both ND and HFD[35]. It is possible that Sirt1 may exert multiple functions in slowing down fat storage in addition to its role in fat mobilization[36].

Of note, the higher EE in *Tip60^{SA/SA}* mice obviously contributes to their resistance to obesity. However, considering that fat oxidation rate is not altered in cultured cells with either Tip60 depletion or inhibition (Fig. 2i), the increased fat catabolism of *Tip60^{SA/SA}* mice evidenced by the elevated EE, $VO_2$ and reduced RER (Fig. 1q and Supplementary Fig. 1j,k) could be attributed to tissue crosstalk in the organismal level. For instance, attenuated lipid synthesis from fatty acids in adipose tissues could lead to complementary increased fatty acid oxidation in metabolically active tissues such as the skeletal muscles in *Tip60^{SA/SA}* mice, which express higher level of *Cpt1β* (Supplementary Fig. 1m). Consistent with these observations, increased fat catabolism is commonly found in mouse models with loss of enzymes in TAG synthesis[8, 19, 20].

Similar to fat catabolism, Tip60-Ser86 phosphorylation was higher in eWAT of WT HFD-fed mice compared with ND-fed mice (Fig. 3g), but did not change in response to OA in 3T3-L1 adipocytes (Fig. 3d and Fig. 6e). Since phosphorylation of Tip60 at Ser86 strongly augments its acetyltransferase activity, it is possible that reduced AKT/PKB signaling during HFD caused by ceramides[37], adipokines[38] or other signaling lipid factors deinhibits phosphorylation of Tip60 by GSK3[15, 18, 25] and in turn cooperate with fatty acids to induce lipin 1 acetylation and TAG synthesis.

Additional physiological evidence for a critical role of Tip60 in optimal TAG production is the markedly reduced milk lipid content in postpartum *Tip60^{SA/SA}* females (Fig. 2b, c). We have also provided evidence that the functional role of Tip60-dependent lipin1 acetylation in TAG accumulation is conserved in *S. cerevisiae*, suggesting that this regulatory mechanism is fundamental to cellular lipid metabolism. Of note, in *S. cerevisiae*, the activity of a second PAP, App1p, may account for the TAG synthesis when Pah1p is not active due to diminished acetylation (Fig. 7i)[35].

It remains of interest to elucidate how acetylation regulates the translocation of lipin 1 to ER. Based on disorder prediction using MobiDB, we observed that both K425 and K595 are localized in structurally flexible regions of lipin 1 (aa 421–456 and aa 566–616), the structural properties of which have been reported to be significantly changed upon post-translational modifications[39]. It is thus possible that the acetylation of lipin 1 on these sites may induce disorder-to-order transition to reshape lipin 1 into an ER-binding structure, or a conformation that could recruit yet unidentified factor(s) that mediates lipin 1 translocation onto ER. Since the intrinsic PAP activity of lipin 1 is not affected by acetylation in a cell-free system with PA provided in excess, and lipin 1 could bind to artificial liposomes that consist of only PC and PA in an acetylation-independent manner (Supplementary Fig. 4a,d), it is likely that some additional as yet unidentified factor(s) may facilitate acetylated lipin 1 to anchor onto ER in vivo. In addition, it is also worth investigating whether there exists a cycling mechanism that follows ER-anchoring of lipin 1 and the post-translational modifications of lipin 1 required to complete the cycling. In sum, we have uncovered an evolutionarily conserved mechanism in the regulation of lipid synthesis

and storage centered on the reciprocal regulation of lipin 1 acetylation and deacetylation by Tip60 and Sirt1. These findings may thus open new avenues for the treatment of obesity and its associated complications.

## Methods

**General data reports.** No statistical methods were used to predetermine sample size. The experiments were not randomized. The investigators were not blinded to allocation during experiments and outcome assessment.

**Animal studies.** All animal procedures were performed with an approved protocol from the Institutional Animal Care and Use Committee of Xiamen University. Mice were housed in a temperature-controlled environment under a 12 h light/dark cycle with free access to water and standard rodent chow diet. Male C57BL/6J 6-week-old mice were used for the experiments unless otherwise stated. For HFD study, 6-week-old mice were transferred to a 60% fat HFD. Triacylglycerol (Wako, Cat. 290–63701) levels and plasma insulin (ALPCO, 80-INSMSU-E01) were measured according to the manufacturer's instructions. Blood glucose values were determined using OneTouch UltraVue automatic glucometers (Johnson & Johnson). Food intake was collected for 4 consecutive weeks during HFD. For calorimetry experiments, mice fed HFD for a week were individually housed for another 3 days. Oxygen consumption, carbon dioxide production, physical activity and RER were then simultaneously measured for 3–4 days in metabolic chambers with a LabMaster system (TSE Systems). For histology, mouse tissues were fixed in 4% paraformaldehyde and paraffin embedded. Sections of 5 μm were used for hematoxylin and eosin (H&E) staining. For glucose tolerance test, mice were fasted for 16 h and then injected with glucose (2 g/kg, intraperitoneally). For insulin tolerance test, mice were fasted for 3 h and injected with insulin (1 U/kg, intraperitoneally). Milk collection and staining of milk fat globules with BODIPY 493/503 (Invitrogen, Cat. D3922) was performed as described previously[40]. Briefly, female mice were separated from their newborns at day 2 of lactation for 3–4 h and injected with 10 units of oxytocin (Selleck, P1029). At 10 min after the injection, milk was manually collected from the fourth pair of mammary glands for further analysis. For immunofluorescence staining, the milk fat globules were stained by BODIPY 493/503 for 0.5 h, and directly observed by using a confocal microscope (Zeiss LSM 780).

**Tip60^{S86A} knock-in mice.** The targeting vector was constructed by replacing exon 3 to exon 6 of the *Tip60* gene with the modified genomic sequence changing Ser[86] to alanine in company with a neomycin selection cassette, generating the mutant Tip60-S86A. The linearized targeting vector was electroporated into embryonic stem cells. The neomycin-resistant clones were selected. Chimeric mice were generated by microinjection of targeted embryonic stem cells into E3.5 blastocysts. Chimeras derived from embryonic stem cells were bred to C57BL/6J. The genotypes of the resulting progenies were determined by PCR analysis using specific primers as follows: primer F1, 5′-GCCCCAGCCTCGGTTTTCCC-3′; primer R1, 5′-ACTGCCCTACGGGCTGACCC-3′; primer M1, 5′-TCAAGCTGATCCCCCGGGCT-3′.

**Plasmid constructs.** Full-length complementary DNAs (cDNAs) encoding human TIP60, lipin 1, GK, GPAT1, AGPAT1, MGAT1, MGAT2, DGAT1, DGAT2, SIRT1, SIRT2, SIRT3, SIRT4, SIRT5, SIRT6, SIRT7, CTDNEP and NEP1-R1 were obtained by PCR using human cDNA. Nucleotide sequences encoding yeast PAH1, ESA1, GPT2, SCT1, AYR1, SLC1, LRO1 and DGA1 were amplified from yeast total DNA. Point mutations of TIP60, lipin 1, SIRT1 and PAH1 were performed by a PCR-based site-directed mutagenesis method using PrimeSTAR polymerase (Takara). Expression plasmids for various proteins were constructed in the pcDNA3.3 vector for transfection or in pBOBI vector for lentivirus infection. All PCR products were verified by sequencing. The details for the primer sequences used for point mutations and deletion mutations are available upon request.

**WAT explant isolation and lipid synthesis assays.** Fresh epididymal WAT was isolated from WT or *Tip60^{SA/SA}* mice and sliced into small pieces (~10 mg each). The adipose tissue explants were then kept in 24-well plates (3 pieces per well) in an assay buffer (129 mM NaCl, 5 mM NaHCO_3, 4.8 mM KCl, 1.2 mM KH_2PO_4, 2 mM CaCl_2, 1.2 mM MgSO_4, 10 mM HEPES (pH 7.5), 6 mM glucose and 1% fatty acid-free bovine serum albumin (BSA)) in a humidified incubator containing 5% $CO_2$ at 37 °C for 1 h. The adipose tissue explants were then removed to fresh assay buffer supplemented with 0.25 mM BSA-bound oleic acid and 3 μCi of [9, 10-³H]-oleic acid/ml and cultured for 1.5 h. Total lipids were then extracted from the tissue explants as previously described[41] and separated by thin layer chromatography (TLC) on silica layers (Macherey-Nagel, TLC Plates Polygram SIL G 805013). TAG and DAG were separated using hexane/diethylether/acetic acid (80:20:1, v/v) as solvent. PC was separated using chloroform/methanol/water (65:25:4, v/v) as solvent.

**Cell culture and transfection**. HEK293T, HEK293 and 3T3-L1 cells were obtained from ATCC. MEFs were established from mouse embryos (embryonic day 13.5). HEK293T, HEK293, 3T3-L1 and MEFs were maintained in Dulbecco's modified Eagle's medium (DMEM) supplemented with 10% fetal bovine serum (FBS), 2 mM L-glutamine, 100 IU penicillin and 100 μg/ml streptomycin in a humidified incubator containing 5% $CO_2$ at 37 °C. No cell lines used in this work were listed in the ICLAC database. All cell lines were verified to be free of mycoplasma contamination and were authenticated by short tandem repeat (STR) profiling. Cells were treated with 1 μM TSA and 5 mM NAM for 6 h before harvest to inhibit deacetylase activity in experiments in which lipin 1 acetylation needed to be maintained, except for Fig. 3a, b, d, e. For OA treatment, OA was conjugated to fatty acid-free BSA before use as described previously[42], and cells were serum-starved overnight and treated with 0.25 mM OA in high glucose serum-free DMEM. Polyethylenimine at a final concentration of 10 μM was used for transfection of HEK293T or HEK293 cells. Cells were harvested 16–20 h after transfection with a lysis buffer (20 mM Tris-HCl (pH 7.4), 150 mM NaCl, 1 mM EDTA, 1 mM EGTA, 1% Triton, 2.5 mM sodium pyrophosphate, 1 mM β-glycerolphosphate, 1 mM $Na_3VO_4$, 2 μg/ml leupeptin, 1 mM phenylmethylsulfonyl fluoride (PMSF) and deacetylase inhibitors (2 μM TSA and 10 mM NAM)) and immunoprecipitated as previously described[15].

**Adipogenic differentiation**. 3T3-L1 preadipocytes or the primary WT or *Tip60*[SA/SA] MEFs (embryonic day 13.5) within passage 3 were seeded in cell culture plates and propagated to confluence. After 2 days, which was designated as day 0, differentiation was initiated using DMEM supplemented with 10% FBS, 5 μg/ml insulin, 1 μM dexamethasone (Sigma, Cat. D4902), 0.5 mM 3-Isobutyl-1-methylxanthine (IBMX) (Sigma, Cat. I7018) and 10 μM rosiglitazone (Sigma, Cat. R2408). At 2 days after initiation, the culture medium was replaced with a maintenance medium (DMEM supplemented with 10% FBS and 5 μg/ml insulin). Fresh maintenance medium was replaced every 2 days thereafter.

**Fatty acid oxidation measurement**. To determine the fatty acid oxidation rates of adipocytes, cells were cultured in 6-well plates and washed once with pre-warmed PBS. Cells were then incubated in 1 ml reaction buffer ([119 mM NaCl, 5 mM KCl, 2.6 mM $KH_2PO_4$, 2.6 mM $MgSO_4$, 2 mM $CaCl_2$, 25 mM $NaHCO_3$, 10 mM HEPES (pH 7.4), 1 mM BSA-bound oleic acid and 0.8 μCi/ml [9,10-$^3$H(N)]-oleic acid) at 37 °C for 1 h. After that, 480 μl of supernatant was transferred and mixed with 192 μl of 1.3 M perchloric acid. The precipitated protein was then removed by centrifugation at 10,000 rpm for 30 s, and 500 μl supernatant was mixed with 4 ml of scintillation liquid to count the $^3$H radioactivity. Controls without the addition of cells were also contained within each experiment to ensure that no more than 20% of labeled oleic acid was oxidized during the assay.

**RNA interference and reconstitution in 3T3-L1 adipocytes**. The shRNA-mediated knockdown of *Tip60*, *Sirt1* and *Lpin1* in differentiated 3T3-L1 adipocytes was achieved by lentivirus-mediated infection, as previously described[43]. The lentivirus-based vector pLL3.7 was used for expression of shRNAs. The 19-nucleotide sequences for siRNAs to mouse *Tip60* are 5'-GGCTGGACTTAAAGA AGAT-3' (#1) and 5'-GGACTTAAAGAAGATCCAA-3' (#2), to mouse *Sirt1* are 5'-GCACTCAATTCCAAGTTCTA-3' (#1) and 5'-GCACCGATCCTCGAACAAT-3' (#2), to mouse *Lpin1* is 5'-GGAACTCTGTAGACAGAAT-3'. pLL3.7-Renilla was used to express control shRNA (5'-GTAGCGCGGTGTATTATAC-3'). Lentiviruses for infection were packaged in HEK293T cells with the lentivirus expression plasmid and packaging plasmids using TurboFect transfection reagent (Thermo Scientific, Cat. R0531). Viruses collected 48–72 h after transfection were filtered through 0.22 μm filters and then concentrated by ultracentrifugation at 50,000 × g for 2 h and subsequently purified with 20% sucrose by ultracentrifugation at 46,000 × g for 2 h. The infection of 3T3-L1 adipocytes (8 days after differentiation) was carried out by adding lentivirus into the cell culture medium at the multiplicity of infection of 120 and 10 μg/ml polybrene (hexadimethrine bromide) (Sigma, Cat. H9268). The plates containing cells and viruses were then centrifuged at 2500 rpm for 30 min. At 6 h after the infection, the medium was changed. For reconstitution of Flag-tagged lipin 1 or its mutants, 3T3-L1 adipocytes expressing shRNA targeting *Lpin1* were further infected with lentivirus expressing vector, Flag-tagged lipin 1 or its mutants packaged in HEK293T cells, the related plasmids were constructed based on the pBOBI vector. All experiments were carried out at least 48 h after infection.

**Antibodies and drugs**. Antibodies for TIP60 and phospho-Ser[86]-TIP60 were generated as described previously[16]. The antibodies that specifically recognize acetylated or phosphorylated lipin 1 were raised by immunizing rabbits with synthetic peptides: CNGDPSGLA(acK)HASDNG for Ac-Lys[425]-lipin 1, CQLSLATRV(acK)HESSS for Ac-Lys[595]-lipin 1 and MHLAT(pS)PILSEGASC for phospho-Ser[106]-lipin 1. Antibodies to lipin 1 (Cat. #14906, 1:1000), acetylated lysine (Cat. #9681, 1:1000, for detecting lipin 1 acetylation; Cat. #9441, 1:1000, for detecting PAH1 acetylation), β-tubulin (Cat. #2128, 1:1000), ACC (Cat. #3662, 1:1000), ATGL (Cat. #2138, 1:1000), HSL (Cat. #4107, 1:1000), SCD1 (Cat. #2794, 1:1000), C/EBPβ (Cat. #3087, 1:1000), FASN (Cat. #3180, 1:1000), PPARγ (Cat. #2443, 1:1000) and SIRT1 (Cat. #8469, 1:1000) were obtained from Cell Signaling

Technology. Antibodies to Myc (clone number 9E10, 1:500), HA (clone number F-7 and Y-11, 1:1000), lipin 1 (clone number B-12 and H-120, 1:1000), CD36 (clone number H-300, 1:1000), perilipin (clone number H-300, 1:1000) and UCP1 (clone number C-17, 1:1000) were purchased from Santa Cruz Biotechnology. Antibodies to Flag (Cat. F2555 and SAB4200071, 1:5000), Actin (Cat. A1978, 1:5000) and anti-Flag beads (Cat. M8823) were purchased from Sigma. Antibody to Fabp4 (Cat. 12802-1-AP, 1:1000), Acsl5 (Cat. 15708-1-AP, 1:1000) and Calnexin (Cat. 10427-2-AP, 1:1000) was purchased from Proteintech. OA (Cat. O1383), TSA (Cat. T1952), Acetyl-CoA (Cat. A2181), ß-Nicotinamide adenine dunucleotide (NAD$^+$) (Cat. N1511) and NAM (Cat. N0636) were purchased from Sigma. Antibody to yeast PGK1 (Cat. 459250, 1:5000) was from invitrogen. MG149 (Cat. S7476, used final concentration at 67 μM) and EX527 (Cat. S1541, used final concentration at 2 μM) were purchased from Selleckchem.

**RNA isolation and RT-PCR**. Total RNA was isolated and reverse transcription was performed from SVFs-derived adipocytes or skeletal muscle of WT or *Tip60*[SA/SA] mice. The resulting cDNA was diluted in DNase-free water followed by quantification by real-time PCR (RT-PCR). mRNA transcript levels were measured using Applied Biosystems 7900HT Sequence Detection System v2.3 software. All data are expressed as the ratio between the expression of target gene to the housekeeping gene Actin. The following primers were used for quantitative real-time PCR: Cd36-forward: 5'-TCTTGGCTACAGCAAGGCCAGATA-3'; Cd36-reverse: 5'-AGC-TATGCATGGAACATGACG-3'; Fabp4-forward: 5'-GAAAACGAGATGGTGA-CAAGC-3'; Fabp4-reverse, 5'-TTGTGGAAGTCACGCCTTT-3'; Fsp27-forward: 5'-GTGTCCACTTGTGCCGTCTT-3'; Fsp27-reverse: 5'-CTCGCTTGGTTGTCTTGATT-3'; Lpl-forward: 5'-GCCCTA-CAAAGTGTTCCATTACC-3'; Lpl-reverse: 5'-TCATGAGCAGTTCTCC-GATGTCCA-3' Cpt1ß-forward: 5'-GTCGCTTCTTCAAGGTCTGG-3' Cpt1ß-reverse: 5'-AAGAAAGCAGCACGTTCGAT-3'; Actin-forward: 5'-TTGTAAC-CAACTGGGACGATATGG-3'; Actin-reverse: 5'-CGACCAGAGGCATA-CAGGGACAAC-3'.

**Recombinant lipin 1 and in vitro acetylation assay**. Expression and purification of His-tagged lipin 1 and Tip60 were performed as previously described[15, 30]. Briefly, lipin 1 and Tip60 cDNA were cloned into pET-28a vector. Protein expression was performed in the strain BL21 *Escherichia coli* cells. The *E. coli* were incubated for 2 h with 1 mM isopropyl-D-thiogalactopyranoside to induce the expression of His-tagged lipin 1 and Tip60. Ni$^{2+}$-NTA-agarose was used for the purification of His-tagged proteins. For the acetylation reaction, purified His-lipin 1 (about 2 μg) or immunopurified Flagged TAG synthetic enzymes (expressed in HEK293T cells) were incubated with His-TIP60 (about 1 μg) or immunopurified Flagged TIP60 (expressed in HEK293T cells) in 40 μl of reaction buffer containing 20 mM Tris-HCl (pH 8.0), 20% glycerol, 100 mM KCl, 1 mM dithiothreitol (DTT) and 0.2 mM EDTA with or without 100 μM acetyl-CoA. After incubation for 1 h at 30 °C, the reaction was stopped by addition of 10 μl of 5× SDS sample buffer. Samples were then subjected to sodium dodecyl sulfate–polyacrylamide gel electrophoresis (SDS–PAGE) and western blotting.

**In vitro deacetylation assay**. For the deacetylation reaction, Flag-tagged SIRT1 or its inactive mutant was transfected into HEK293T cells. Cells were lysed 18 h after transfection. Flag antibody-conjugated beads were then used for immunoprecipitation for 3 h at 4 °C with end-to-end rotation. The beads were then washed three times with lysis buffer and twice with 25 mM Tris-HCl (pH 7.5) buffer. The Flag-tagged SIRT1 were eluted with 3× Flag peptide (Sigma, Cat. F3290) in 25 mM Tris-HCl (pH 7.5) buffer containing 150 mM NaCl and protease inhibitors. The acetylated lipin 1 (prepared by immuneprecipitating the Flag-tagged WT-lipin 1 co-expressed with WT-TIP60 in HEK293T cells) was then incubated with immunopurified Flag-tagged SIRT1 or its inactive mutant in 40 μl of reaction buffer containing 50 mM Tris-HCl (pH 9.0), 2 mM $MgCl_2$, 50 mM NaCl, 0.5 mM DTT and 0.2 mM PMSF with or without 1.5 mM NAD$^+$. After incubation for 1 h at 30 °C, the reaction was stopped by addition of 10 μl of 5× SDS sample buffer. Samples were then subjected to SDS–PAGE and western blotting.

**PAP activity measurement**. HEK293 cells were transfected either with vector or Flag-tagged lipin 1 with or without TIP60. Cells were lysed 18 h after transfection. Flag antibody-conjugated beads were then used for immunoprecipitation for 3 h at 4 °C with end-to-end rotation. The beads were then washed three times with lysis buffer and twice with 25 mM Tris-HCl (pH 7.5) buffer. The Flag-tagged lipin 1 was then eluted with 3× Flag peptide in 25 mM Tris-HCl (pH 7.5) buffer containing 150 mM NaCl and protease inhibitors. The eluents were further used for measurement of the PAP activity as described previously[12, 44], with slight modifications. Briefly, 5–20 μl of each eluent was added to reaction mixtures (80 μl) containing a final concentration of 0.1 M Tris/maleate (pH 6.9), 10 mM $MgCl_2$, 0.2% fatty acid-free BSA, 0.2 mM [$^{14}$C] phosphatidic acid (American Radiolabeled Chemicals, Cat. ARC1574), 1 mM DTT and 1× EDTA-free protease inhibitor. After 20 min at 37 °C, the assay reactions were terminated by adding 1 ml of a chloroform/methanol mixture (19:1, v/v) containing 0.8% olive oil and vortexing well. Then, 500 mg of activated $Al_2O_3$ was added to each tube and capped securely. After three cycles of vortexing for 30 s and being left undisturbed for 10 min,

samples were centrifuged at $10,000 \times g$ for 10 min, 350 µl of supernatants was used for scintillation to count the $^{14}$C radioactivity and vector transfected values were subtracted. Activity measurements were normalized to protein present in the starting lysate determined by the Bradford assay.

**Subcellular fractionation.** 3T3-L1 adipocytes were fractionated as described previously[34]. In brief, for each treatment condition, a 10 cm diameter plate of cells were rinsed in iced PBS once and resuspended in 1 ml of Buffer A (0.25 M sucrose, 50 mM NaF, 1 mM EDTA, and 50 mM Tris-HCl, pH 7.4) supplemented with 1 mM PMSF, 2 µg/ml leupeptin and 1 mM $Na_3VO_4$. The homogenized cells were then mechanically broken by spraying 8 times through a 0.45 µm needle and centrifuged at $100 \times g$ for 10 min to obtain the nucleus fraction. After that, the supernatants were centrifuged at $16,000 \times g$ for 20 min, and the pellets containing plasma membranes and mitochondria were retained. Finally, the supernatants were centrifuged at $175,000 \times g$ for 1 h. The high speed supernatants containing cytosol fraction were collected, and the pellets containing microsomal fraction were obtained.

**Immunofluorescence analysis of 3T3-L1 adipocytes.** 3T3-L1 adipocytes expressing control shRNA or shRNA (#1) targeting *Tip60* were serum-starved overnight and treated with BSA or 0.25 mM OA in high glucose serum-free DMEM for 2 h. Cells were then fixed with 4% paraformaldehyde, rinsed with PBS for two times, permeabilized with 0.1% Triton X-100 (diluted in PBS) for 10 min at 4 °C and then blocked with 5% BSA for 30 min. Cells were rinsed two times with PBS and were incubated with primary antibodies to lipin 1 (Santa, B-12) and Calnexin (Proteintech, Cat. 10427-2-AP) overnight at 4 °C. The cells were then rinsed with PBS three times, and incubated with Alexa-Fluor 488-conjugated and 594-conjugated secondary antibodies for 6 h. Fluorescence images were obtained with a Zeiss 780 confocal microscope and processed with Adobe Photoshop for presentation.

**Liposome preparation and binding assay.** Liposomes consisting of PC and PA were prepared as described previously[27]. To measure the binding of lipin 1 to liposomes, immunopurified Flag-tagged WT-lipin 1 or 2KR mutant co-expressed with or without Tip60 in HEK293T cells were incubated with liposomes consisting of PC or PC with 20 mol% of PA for 20 min at 30 °C. The lipin 1 protein that binds with liposomes was then purified by ultracentrifuge at 4 °C for 20 min at $100,000 \times g$ and analyzed by western blotting.

**Lipid extraction and analysis of PA levels by LC-MASS.** Lipids in eWAT were extracted as described previously[45]. Briefly, frozen eWAT was inactivated by addition of 900 µl of chloroform/methanol (1:2) containing 10% deionized $H_2O$. Samples were then homogenized on an automated bead ruptor (Omni) and centrifuged at 1500 rpm at 4 °C for 1 h. After that, 400 µl of deionized $H_2O$ and 300 µl of chloroform were added. The lower organic phase was transferred to a new tube. A second extraction was carried out via addition of 500 µl of chloroform. The two extractions were pooled into the same tube and dried using SpeedVac (Genevac). Samples were stored at −80 °C until mass spectrometric analysis.

Quality control sample was run at the 1st, 7th and 15th sample in the queue sequence. All reported peaks were identified manually based on desirable peak shapes and signal-to-noise ratios of >3, and with coefficients of variations of less than 12% among replicates of the same biological groups. Qualitative deuterated lipid standards from LIPID MAPS were pre-corrected using suitable quantitative lipid standards from the same lipid class based on molar response prior to their use for quantitation. All mass spectrometric analyses were carried out on a Sciex Exion UPLC with a Sciex 6500 QTRAP Plus.

Separation of individual lipid classes of polar lipids by normal-phase high-performance liquid chromatography was carried out using a Phenomenex Luna 3u silica column (i.d. $150 \times 2.0$ mm) with the following conditions: mobile phase A (chloroform/methanol/ammonium hydroxide, 89.5:10:0.5), phase B (chloroform/methanol/ammonium hydroxide/water, 55:39:0.5:5.5). Multiple reaction monitoring transitions were set up for quantitative analysis of various polar lipids. Individual lipid species were quantified by referencing to spiked internal standards. For all LC/MS analyses, individual peaks were manually examined and only peaks above the limit of quantitation and within the linearity range were used for quantitation. The absolute amounts of all qualitative lipid standards were pre-corrected against quantitative standards prior to their use for quantitative purposes. Details of the analytical protocol have been previously described elsewhere[45, 46].

**Yeast strains and growth conditions.** Most yeast strains were kind gifts from Dr. Li Yu's lab (Tsinghua University) as previously described[33]. The wild-type BY4741 (*MATa his3Δ1 leu2Δ met15Δ ura3Δ*) and deletion derivatives were purchased from Invitrogen. The chromosomally integrated *Faa4-GFP* fusion strains were generated by tagging the *faa4* locus with a *GFP*-tag sequence as well as the selection marker *KanMX6*, as described previously[47]. The *esa1-HA-aid* strain for the rapid depletion of ESA1 was created by using an auxin-based degron system, as described previously[48]. Briefly, a PCR-based one-step auxin-inducible degron (AID) tagging method was performed at the *esa1* locus using a yeast strain in which one copy of the *AtTIR1* gene driven by the constitutive *ADH1* promoter was chromosomally

integrated. All deletion strains and epitope-tagged constructs in this study were verified by sequencing. The genotypes of yeast strains used are listed in Supplementary Table 2.

Yeast cells were grown in YEPD medium (1% yeast extract, 2% peptone, 2% glucose) or in synthetic complete (SC) medium containing 2% glucose at 30 °C as described previously[29]. For selection of yeast cells bearing plasmids or specific modifications on the chromosome together with certain selection markers, appropriate amino acids were omitted from SC medium. For cells bearing the selection marker *KanMX6*, 0.2 mg/ml Geneticin was used. Yeast transformation or analysis of proteins expressed in yeast by western blotting was carried as described previously[33].

For analysis of the lipids in yeast, yeast cells were grown in SC medium to log-phase growth ($OD_{600} = 0.8$–1.0) and treated with or without the auxin indole-3-acetic acid (IAA) (Sigma, Cat. I5148) at 0.5 mM (auxin was specifically used to deplete ESA1 in the strains with *esa1-HA-aid*). The yeast cells were then further cultured for 4 h and either analyzed by staining or by total lipid extraction. Appropriated controls such as wild-type BY4741 strains treated with auxin or *esa1-HA-aid* strains treated with dimethyl sulfoxide (DMSO; used to dissolve IAA) were also included during the experiments. It was found that neither expression of AtTIR1 nor addition of auxin or DMSO affected the cell growth or TAG accumulation in yeast, as previously reported[48]. Whole-cell extracts of yeast cells were obtained as described previously[33].

**Microscopy of yeast strains.** For fluorescent microscopy, the cells were either stained by BODIPY 493/503 (Invitrogen, Cat. D3922) for 0.5 h and observed, or directly observed (for strains expressing the Faa4-GFP) by using a confocal microscope (Zeiss LSM 780). The percentage lipid droplet area per cell was determined by analyzing at least 100 cells for each group.

**Total lipid extraction and TLC for yeast samples.** For total lipid analyses, yeast cells were broken with glass beads by vigorous overtexing for 10 min at 4 °C, and lipids were extracted with chloroform/methanol 2:1 (v/v), as described previously[29]. Neutral lipid analysis was performed by TLC by applying samples onto silica gel plates. TAG was separated by TLC on silica gel plates by using hexane/diethylether/acetic acid (80:20:1, v/v) as the solvent. Lipids were then stained by iodine vapor and quantified.

**Statistical analyses.** For all the data used for statistical analysis, each group of data was analyzed for the criteria for each test before performing the tests (e.g., normal distribution and homogeneity of the variance). The homogeneity of variance was tested by Levene's test. For experiments with only two groups, Student's $t$-test was used for statistical comparisons. Analysis of variance (ANOVA) with Tukey's post test (one-way ANOVA for comparisons between groups, two-way ANOVA for comparisons of magnitude of changes between different groups from different cell lines or treatments) was used to compare values among different experimental groups using SPSS Statistics 17.0 or Graphpad Prism 6 program. Weekly food intake was determined by ANCOVA. No samples or animals were excluded from the analysis. $P < 0.05$ was considered as statistically significant (*) and $P < 0.01$ as highly significant (**).

**Data availability.** The data supporting the conclusions of this work are included within the article, Supplementary Files and available from the corresponding author upon reasonable request.

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

## Acknowledgements

We thank Dr. L. Yu from Tsinghua University for yeast strains, Dr. P. Li and Dr. T. J. Zhao for useful discussions. This work was supported by grants from the National Natural Science Foundation of China (#31690101, #31430094, #31600961 and #31571214) and National Key Research and Development Project of China (2016YFA0502001).

## Author contributions

T.Y.L., L.S., Y.S. and S.-C.L. conceived the project and designed the experiments. T.Y.L., L.S., Y.S., J.L., C.Y., Y.Y. and C.-S.Z. performed experiments and participated in discussion of the results. S.M.L., D.X., L.Z. and G.S. contributed to metabolites measurement. X.L. and X.H. contributed to the generation of antibodies. C.X. contributed to the identification of lipin 1 acetylation sites by LC-MS/MS. S.-Y.L. and K.R. helped with discussion and interpretation of results. T.Y.L., L.S., Y.S., K.R. and S.-C.L. analyzed data and wrote the paper.

## Additional information

**Competing interests:** The authors declare no competing interests.

