## [Peer Review File · Nature Communications]

Reviewers' comments:

Reviewer #1 (Remarks to the Author):

Li and colleagues have generated a TIP60 knockin mouse with diminished catalytic activity. There are several notable phenotypes of the mice, and the authors further explore the diminished adiposity by dissecting mechanism. There are convincing and novel data that lipin 1 is a target of this acetyltransferase and that this modification of the protein affects the activity by regulating its localization. The manuscript is very well written and strong from a technical standpoint, especially with regards to the regulation of lipin 1 acetylation. It remains to be determined how large a role the regulation of lipin 1 plays in the phenotype of TIP60 knockin mice.

The knockin (KI) mice are clearly lean and the adipocytes from these mice have diminished capacity for lipid synthesis *in vitro*. However, it's less clear that this is due to altered lipin 1 acetylation and activity or other effects of reduced TIP60 activity. There are a series of papers by Kalkhoven's group indicating that TIP60 plays a role in adipogenesis by regulating the activity of PPAR γ and that moderately inhibiting the expression of this protein affects differentiation. These papers are not cited. Moreover, the expression of the known TIP60 target genes is also not examined *in vivo* or *in vitro* nor is the phenotype of the adipocytes examined carefully. Can they fully differentiate? Is the expression of lipid uptake and storage genes affected? What are the transcriptional effects of this knockin and do they affect the results?

Related to this as well, the most convincing experiment to show that lipin 1 is involved in the phenotype of the KI mice would be to express lipin 1 protein that would mimic constitutive acetylation by mutating the lysines to glutamine in the KI adipocytes. If this made lipin 1 act as an acetylated protein and overcame the diminished lipogenic phenotype, that would be convincing.

It is not clear how lipin 1 and TIP60 or Sirt1 are able to interact as the acetyltransferase and deacetylase are nuclear. Does this occur in the nucleus and is the nuclear localization affected by acetylation? The K595 residue seems to be the same amino acid that is sumoylated in a previous study and is involved in regulating nuclear localization by that post translational modification.

In addition, it is stated that phosphorylation of lipin 1 affects the "affinity" for TIP60 (Figure 6a data) which is not really shown or tested. The 16SA lipin mutant is supposed to be more nuclear, which could put it in proximity to TIP60 in the nucleus rather than affecting affinity.

Minor:

First paragraph of the Introduction: The term "phosphatidic phosphatase" should be replaced with "phosphatidic acid phosphohydrolase"

Reviewer #2 (Remarks to the Author):

In this study, Li and colleagues have identified a novel regulation of how Tip60 mediates the sensing of fatty acids to stimulate TAG synthesis. Mechanistically, fatty acid induces acetylation of Lipin1 by Tip60, and anchors Lipin 1 onto ER membrane to convert DAG into TAG. They have further mapped the acetylation sites on Lipin 1, identified the deacetylase Sirt1, and investigated the interplay between phosphorylation and acetylation, followed by functional validation in yeast. Overall, this study uncovers an important regulation of lipid metabolism accompanied by comprehensive and definite mechanism.

Q1. Is the suppression of TAG synthesis by Sirt1 dependent on Lipin 1? In Lipin 1 knockdown adipocytes or Tip60 SA mutant adipocyte, does its suppression still persist? In the Lipin 2KR mutant reconstituted 3T3-L1 cells, does Tip60 loss of function (knockdown by shRNA or SA

mutant) still suppress TAG synthesis? These studies will help to clearly establish the proposed model.

Q2. It's quite striking that lipin 1 completely fails to translocate ER membrane in response to fatty acid in the Tip60 KD cells (Figures 5a, 5b), and the Tip60 SA mutant show a lower response in line with its 50% remaining acetyltransferase activity (Figure 3f). There is basal about 50% location of lipin 1 on ER membrane (Figures 5a-c). Does it mean there is another regulatory mechanism accounted for this basal location? Figures 5e-f are supposed to give some hints to this question, however, the remaining response to OA (Fig 5f), although overall lower, could be caused by the leftover WT lipin 1 from knockdown. Therefore, this question remains open from the current evidences.

Q3. The proposed model in Figure 6f seems incomplete. Does Tip60 co-localize with acetylated lipin 1 on ER membrane? What happens after acetylated Lipin 1 translocate to ER membrane? Sirt1 binds to it to deacetylate it and then recycle it to cytosol? At which step is lipin 1 phosphorylated, before or after Sirt1 binding? The 2KR mutant has lower binding affinity to Tip60 (Figure 6a). It seems phosphorylation is an important component for lipin 1 to complete the cycling.

Q4. Does Tip60 S86A mutation cause lower binding affinity to Lipin 1?

Q5. Is the plasma fatty acid level lower in SA mice than in WT mice due to the deficiency of TAG synthesis?

Minor:

Page 8, line 10: "if" should be "whether".

Reviewer #3 (Remarks to the Author):

Major findings:

The authors constructed knock-in mice expressing a Tip60 allele that encodes for alanine instead of serine at position 80. This precludes phosphorylation by GSK3 (and perhaps other kinases) on serine 80. This decreases activation of Tip60 in response to this phosphorylation (described in earlier work). These mice were resistant to adipose expansion and weight gain with concomitant maintenance of insulin sensitivity when placed on a high fat diet. Milk triglyceride production was decreased. A mechanistic explanation was supported by evidence that Tip60 directly interacts with and may acetylate the phosphatidic acid phosphatase, lipin1. Site directed mutagenesis identified two lysine residues in lipin1 that were the targets of Tip60-mediated acetylation. Sirt1 was identified as the deacetylase that may reverse the Tip60-mediated acetylation of lipin1. Fluorescence microscopy supported that lipin1 translocates to the ER upon acetylation. Unphosphorylated lipin1 was a better substrate for Tip60, suggesting that dephosphorylation of lipin 1 by an unnamed phosphatase precedes Tip60 acetylation and subsequent activation of lipin1. Parallel studies in *S. cerevisiae* supported that a homologous regulatory system, via acetylation of the lipin1 homolog, Pah1, exists.

This study provides a physiologically novel mechanism for acetylation regulating phosphatidic acid phosphatase activity and thus, triglyceride synthesis. The experiments presented were exhaustive and thorough with unambiguous, high-quality data throughout. Elucidating novel mechanisms for regulation energy allocation, particularly related to fatty acid allocation and triglyceride synthesis, have clear importance in both basic science and applied disciplines.

Major Point

The discussion would be improved by addressing additional considerations of the data presented.

These considerations include addressing the following questions:

- a) Nem1p - Spo7p have been implicated in the dephosphorylation and subsequent activation of Pah1p (Pascaul, F. 2014). Are there human homologs of these that might be implicated in the pre-acetylation, phosphorylation-mediated regulation of lipin1?
- b) The authors should acknowledge the presence of a second PAP in the yeast genome, APP1 (Chae, M, 2012), and how App1p activity may account for the TAG synthesis when Pah1p is not active due to diminished acetylation.
- c) To extent have heterozygous, Tip60 wt/SA mice been investigated for these phenotypes studied, is there any evidence that heterozygosity confers intermediate phenotypes?
- d) What is a possible mechanism of how oleate / oleic acid influences Tip60 activity?
- e) Does oleate-induced Tip60 activity also likely change chromatin modification?
- f) Why was only one lysine (425) conserved with the acetylation target in yeast Pah1p? Do the locations of the acetylated lysines yield insight to the structure of lipin1 and how it binds to the ER?

Minor points

pg. 2 line23: Glycerol kinase is not commonly considered part of the glycerol 3-phosphate pathway. However, its contribution to proving the pathway with substrate may be a point for regulation and worth mentioning. In that case, glycerol 3 phosphate dehydrogenase should also be mentioned as it is also contributes to the production of glycerol 3 phosphate.

pg. 3 line 18: Data / reference should be provided to support that PAP changes subcellular location in response to changing conditions as opposed to being a peripheral protein constitutively present on the ER surface.

pg. 5 line 5: A nice control would have been a knock-in of a wild-type Tip60 allele so to account for any effects caused by alterations to the Tip60 locus that may have occurred during the insertion of the SA allele. Generating this control would clearly require a good deal of additional effort that would only marginally strengthen the conclusions made.

pg. 6 line 24: "Lipids, primarily TAG, in the milk supply the majority of the nutrients ..." Consider changing "nutrients" to "calories"

pg. 13, line 24 The possible molecular / cellular mechanism(s) of the proposed "tissue crosstalk" bear proposing in some detail.

Syntax

pg. 2, line 25; That sentence seems to unnecessarily extended by using the semi-colon

pg. 4, line 2 "as transcriptional regulators, growing ..." should be "a transcriptional regulator, a growing"

pg. 11, line 16: "investigate" to "investigated"

pg. 14, line 9 "in turn cooperate with fatty acids to regulate lipin 1 acetylation and TAG synthesis" changing "regulate" to "induce" would make that statement more specific.

pg. 43, line 19, 20 "After three cycles of vortexing for 30 s and being left undisturbed for 10 min." - sentence fragment

Point-by-Point Responses

Reviewer #1:

Li and colleagues have generated a TIP60 knockin mouse with diminished catalytic activity. There are several notable phenotypes of the mice, and the authors further explore the diminished adiposity by dissecting mechanism. There are convincing and novel data that lipin 1 is a target of this acetyltransferase and that this modification of the protein affects the activity by regulating its localization. The manuscript is very well written and strong from a technical standpoint, especially with regards to the regulation of lipin 1 acetylation. It remains to be determined how large a role the regulation of lipin 1 plays in the phenotype of TIP60 knockin mice.

We thank you for saying that our data are convincing and novel, and that our manuscript is well written and technically strong. By performing additional experiments, the conclusion that acetylation of lipin 1 plays a major role in the phenotypes of *Tip60^{SA}* knockin mice has now been further strengthened.

1. The knockin (KI) mice are clearly lean and the adipocytes from these mice have diminished capacity for lipid synthesis in vitro. However, it's less clear that this is due to altered lipin 1 acetylation and activity or other effects of reduced TIP60 activity. There are a series of papers by Kalkhoven's group indicating that TIP60 plays a role in adipogenesis by regulating the activity of PPARγ and that moderately inhibiting the expression of this protein affects differentiation. These papers are not cited. Moreover, the expression of the known TIP60 target genes is also not examined in vivo or in vitro nor is the phenotype of the adipocytes examined carefully. Can they fully differentiate? Is the expression of lipid uptake and storage genes affected? What are the transcriptional effects of this knockin and do they affect the results?

We are sorry that we did not cite enough of the papers indicating the role of Tip60 in adipogenesis, despite that the most representative paper by Kalkhoven's group on the role of Tip60 in adipogenesis was included in our previous manuscript (Page 8, line 8). The other papers by Kalkhoven's group on the role of Tip60 in adipogenesis have also now been cited in the revised manuscript (Page 8, line 9). With regard to the effects of *Tip60^{SA}* on adipogenic differentiation, we have provided data showing that primary MEFs from *Tip60^{SA/SA}* mice demonstrated a similar adipogenic capacity compared to that of WT MEFs (Supplementary Fig. 2b). To further address this issue, we used primary adipose stromal vascular fibroblasts (SVFs) from WT and *Tip60^{SA/SA}* mice to test the effect of *Tip60^{SA}* on adipocyte differentiation. We found that SVFs from both WT and *Tip60^{SA/SA}* mice can fully differentiate to adipocytes (Figure R1). Meanwhile, the expression of multiple known Tip60 target genes (e.g. *Fabp4*, *Cd36*, *Fsp27*, *Lpl* by regulating PPARγ), and genes related to lipid uptake and storage (e.g. *FASN*, *Acc*, *Acl5*, *SCD1*, *perilipin1*, *ATGL*, *HSL*) was not affected by *Tip60^{SA}* knockin, as revealed by western blotting or RT-PCR (Figure R2 and Supplementary Fig. 2c, d). Together, these data indicate that the decreased lipogenic phenotype in *Tip60^{SA}*-knockin mice is probably not caused by altered adipogenesis or transcriptional regulations, but most likely results from our proposed post-transcriptional regulatory mechanism.

Figure R1. Primary adipose stromal vascular fibroblasts (SVFs) from WT and *Tip60*^{SA/SA} mice demonstrated similar ability in adipocyte differentiation. Representative photographs of WT and *Tip60*^{SA/SA} stromal vascular fibroblasts (SVFs) after 8 days of adipogenic differentiation induction.

Figure R2. Expression of lipid metabolism-related genes in adipocytes derived from WT and *Tip60*^{SA/SA} SVFs. Cell lysates of WT and *Tip60*^{SA/SA} SVFs after 8 days of adipogenic differentiation induction were analysed by immunoblotting (left). RT-PCR analysis of gene expressions in WT and *Tip60*^{SA/SA} SVFs after 8 days of differentiation induction (right, $n = 8$ mice for each group). Error bars denote SEM.

2. *Related to this as well, the most convincing experiment to show that lipin 1 is involved in the phenotype of the KI mice would be to express lipin 1 protein that would mimic constitutive acetylation by mutating the lysines to glutamine in the KI adipocytes. If this made lipin 1 act as an acetylated protein and overcame the diminished lipogenic phenotype, that would be convincing.*

Following your instruction, we examined the TAG/DAG synthesis rates of KI adipocytes expressing WT-lipin 1, acetylation-defective (2KR) or acetylation mimetic (2KQ) mutant of lipin 1. It was found that while KI adipocytes expressing WT-lipin 1 demonstrated decreased TAG/DAG synthesis rate than that of WT adipocytes, similar TAG/DAG synthesis rates were detected in KI adipocytes expressing 2KR- or 2KQ-lipin 1 compared to those in the corresponding WT adipocytes (Figure R3 and Fig. 6f). These results indicate that *Tip60*-mediated lipin 1 acetylation plays a major role in determining the differences between WT and KI adipocytes in TAG/DAG synthesis. We have now added this part to the revised manuscript (Page 13, line 7).

Figure R3. Tip60-mediated lipin 1 acetylation plays a major role in determining the differences between WT and *Tip60*^{SA/SA} adipocytes in TAG/DAG synthesis. TAG/DAG production rates of WT or *Tip60*^{SA/SA} adipocytes expressing shRNA targeting *Lpin1* with reintroduction of WT-lipin 1, acetylation-defective (2KR) or acetylated mimetic (2KQ) mutant of lipin 1 ($n = 4$ experiments). Error bars denote SEM. Statistical analysis was performed by ANOVA followed by Tukey ($*P < 0.05$; N.S., not significant).

3. *It is not clear how lipin 1 and TIP60 or Sirt1 are able to interact as the acetyltransferase and deacetylase are nuclear. Does this occur in the nucleus and is the nuclear localization affected by acetylation? The K595 residue seems to be the same amino acid that is sumoylated in a previous study and is involved in regulating nuclear localization by that post translational modification.*

In order to further clarify this issue, we analyzed the localization of Tip60 and Sirt1 in 3T3-L1 adipocytes with or without oleic acid (OA) treatment. In line with previous reports (*Sapountzi et al., Int J Biochem Cell Biol. 2006; Byles et al, Int J Biol Sci. 2010*), while Tip60 and Sirt1 are mainly localized in the nucleus, certain amounts of Tip60 and Sirt1 are also present in the cytoplasm (Figure R4), which is consistent with our cell fractionation results (Figure R5). Meanwhile, we found that the acetylation-defective mutant of lipin 1 (K425/K595R-lipin 1) demonstrated a similar nuclear localization ratio to WT-lipin 1 under both BSA and OA treatment conditions (Figure R6, left), indicating that the nuclear localization of lipin 1 is likely not affected by Tip60-mediated acetylation. As to the sumoylation of K595, we found that while double mutation of the two sumoylation sites (K565/K595) decreased lipin 1 nuclear localization, single mutation of K595 to arginine didn't have such an effect. Moreover, knockdown of *Tip60* by shRNAs didn't affect the nuclear localization of lipin 1 (Figure R6, right). These results indicate that the acetylation statuses of lipin 1 don't affect its nuclear localization.

Figure R4. Subcellular localization of Tip60 and Sirt1 in 3T3-L1 adipocytes with or without OA treatment. 3T3-L1 adipocytes were treated with BSA or OA for 2 h and analysed by immunofluorescence.

Figure R5. Nuclear and cytosolic localizations of Tip60 and Sirt1 in 3T3-L1 adipocytes. Western blotting of proteins in subcellular fractions of 3T3-L1 adipocytes with or without OA for 2 h. TCL, total cell lysate; Lamin B, a nucleus marker; Calnexin, microsomal marker; β -tubulin, cytosol marker.

Figure R6. Lipin 1 nuclear localization is not affected by Tip60-mediated acetylation. Western blotting of proteins in the nuclear fraction of 3T3-L1 adipocytes expressing Flag-tagged WT or mutants of lipin 1 treated with or without OA for 2 h (left). Western blotting of proteins of 3T3-L1 cells expressing control shRNA (ctrl) or shRNAs targeting *Tip60* (right). Lamin B is a nuclear (Nuc) marker. TCL, total cell lysate.

4. In addition, it is stated that phosphorylation of lipin 1 affects the “affinity” for TIP60 (Figure 6a data) which is not really shown or tested. The 16SA lipin mutant is supposed to be more nuclear, which could put it in proximity to TIP60 in the nucleus

rather than affecting affinity.

Thank you for your comment. To clarify this issue, we used a “cell-free system” to determine the affinity between Tip60 and different lipin 1 mutants to exclude the potential influence on the intracellular localization of lipin 1. In this system, Flag-tagged WT-lipin 1 and its mutants were expressed in HEK293T cells and immuno-purified using Flag antibody-conjugated beads. Bacterially expressed His-tagged Tip60 was then added and incubated with the purified lipin 1. After incubation, the beads were then washed and the amount of Tip60 pulled down by lipin 1 was determined by western blotting. The results indicate that compared with WT-lipin 1, the dephosphorylated form of lipin 1 (16SA) indeed showed increased affinity for Tip60 (Figure R6). However, we totally agree with you that the subcellular localization of lipin 1 may also affect its interaction with Tip60 in intact cells.

Figure R7. Interaction of Tip60 and lipin 1 *in vitro*. Flag-tagged WT-lipin 1 and its mutants were expressed in HEK293T cells and immuno-purified using Flag antibody-conjugated beads. Bacterially expressed His-tagged Tip60 was then added and incubated with the purified lipin 1. After 1 h incubation, the beads were washed three times and the amount of Tip60 pulled down by lipin 1 was determined by western blotting.

Minor:

First paragraph of the Introduction: The term “phosphatidic phosphatase” should be replaced with “phosphatidic acid phosphohydrolase”

We have changed this in our revised manuscript (Page 2, line 24).

We thank you again for your constructive comments and patience in reading our manuscript. We hope that the above point-by-point responses have addressed all of your concerns.

Reviewer #2:

In this study, Li and colleagues have identified a novel regulation of how Tip60 mediates the sensing of fatty acids to stimulate TAG synthesis. Mechanistically, fatty acid induces acetylation of Lipin1 by Tip60, and anchors Lipin 1 onto ER membrane to convert DAG into TAG. They have further mapped the acetylation sites on Lipin 1, identified the deacetylase Sirt1, and investigated the interplay between phosphorylation and acetylation, followed by functional validation in yeast. Overall, this study uncovers an important regulation of lipid metabolism accompanied by comprehensive and definite mechanism.

We thank you for your comments that our work is novel and has uncovered an important regulation of lipid metabolism accompanied with a comprehensive and definite mechanism.

Q1. Is the suppression of TAG synthesis by Sirt1 dependent on Lipin 1? In Lipin 1 knockdown adipocytes or Tip60 SA mutant adipocyte, does its suppression still persist? In the Lipin 2KR mutant reconstituted 3T3-L1 cells, does Tip60 loss of function (knockdown by shRNA or SA mutant) still suppress TAG synthesis? These studies will help to clearly establish the proposed model.

To clarify whether suppression of TAG synthesis by Sirt1 is mediated by lipin 1, we treated the 3T3-L1 adipocytes expressing either control shRNA or shRNA targeting *Lpin1* with Sirt1 inhibitor EX527. It was found that inhibition of Sirt1 could no longer increase TAG/DAG synthesis in cells expressing shRNA targeting *Lpin1* (Figure R1 and Fig. 4h), indicating that the suppression of TAG synthesis by Sirt1 is indeed dependent on lipin 1.

Figure R1. Suppression of DAG/TAG synthesis by Sirt1 is mediated by lipin 1. TAG, DAG and PC synthesis rates of 3T3-L1 adipocytes expressing control shRNA (ctrl) or shRNA targeting *Lpin1* with or without EX527 treatment ($n = 4$ experiments). Error bars denote SEM. Statistical analysis was performed by ANOVA followed by Tukey ($*P < 0.05$; N.S., not significant).

Next, we knocked down *Lpin1* in either WT or *Tip60*^{SA/SA} adipocytes and analysed the rates of TAG/DAG synthesis. It was found that knockdown of *Lpin1* strongly diminished the differences of TAG/DAG synthesis between WT and *Tip60*^{SA/SA} adipocytes (Figure R2 and Fig. 4i), indicating that Tip60 regulates TAG/DAG

synthesis through lipin 1.

Figure R2. Knockdown of *Lpin1* diminishes the differences of TAG/DAG synthesis between WT and *Tip60*^{SA/SA} adipocytes. TAG/DAG synthesis rates of WT or KI adipocytes expressing control shRNA (ctrl) or shRNA targeting *Lpin1* ($n = 4$ experiments). Error bars denote SEM. Statistical analysis was performed by ANOVA followed by Tukey ($*P < 0.05$; N.S., not significant).

Finally, we examined the TAG/DAG production rate of *Tip60*^{SA/SA} adipocytes expressing WT-lipin 1, acetylation-defective (2KR) or acetylated mimetic (2KQ) mutant of lipin 1. It was found that while *Tip60*^{SA/SA} adipocytes expressing WT-lipin 1 demonstrated decreased TAG/DAG synthesis rate compared to that in WT adipocytes, the TAG/DAG synthesis rates in *Tip60*^{SA/SA} adipocytes expressing 2KR or 2KQ forms of lipin 1 were similar to those in WT adipocytes (Figure R3 and Fig. 6f), indicating that Tip60-mediated lipin 1 acetylation plays a major role in determining the differences of TAG/DAG synthesis between WT and *Tip60*^{SA/SA} adipocytes.

We have now added these parts to the revised manuscript (Page 10, line 16; Page 11, line 1; Page 13, line 7).

Figure R3. Tip60-mediated lipin 1 acetylation plays a major role in determining the differences between WT and *Tip60*^{SA/SA} adipocytes in TAG/DAG synthesis. TAG/DAG production rates of WT or *Tip60*^{SA/SA} adipocytes expressing shRNA targeting *Lpin1* with reintroduction of WT-lipin 1, acetylation-defective (2KR) or acetylated mimetic (2KQ) mutant of lipin 1 ($n = 4$ experiments). Error bars denote SEM. Statistical analysis was performed by ANOVA followed by Tukey ($*P < 0.05$; N.S., not significant).

Q2. It's quite striking that lipin 1 completely fails to translocate ER membrane in

response to fatty acid in the Tip60 KD cells (Figures 5a, 5b), and the Tip60 SA mutant show a lower response in line with its 50% remaining acetyltransferase activity (Figure 3f). There is basal about 50% location of lipin 1 on ER membrane (Figures 5a-c). Does it mean there is another regulatory mechanism accounted for this basal location? Figures 5e-f are supposed to give some hints to this question, however, the remaining response to OA (Fig 5f), although overall lower, could be caused by the leftover WT lipin 1 from knockdown. Therefore, this question remains open from the current evidences.

We totally agree with you that some other mechanisms may be responsible for the basal ER localization of lipin 1. In fact, consistent with another report (Eaton et al., J Biol Chem. 2013), we found that lipin 1 could bind to artificial liposomes that consist of only phosphatidylcholine (PC) and phosphatidic acid (PA) *in vitro* in an acetylation-independent manner (Figure R4 and Supplementary Fig. 4d), which could contribute to the basal localization of lipin 1 on ER. We have included this part in our revised manuscript (Page 12, line 3).

Figure R4. Binding of lipin 1 to artificial liposomes. Immuno-purified Flag-tagged WT-lipin 1 or 2KR mutant co-expressed with or without Tip60 in HEK293T cells were incubated with liposomes consisting of phosphatidylcholine (PC) or PC with 20 mol% of phosphatidic acid (PA) (PC/PA) for 20 min. The lipin 1 protein binds with liposomes (Myc) were purified by ultracentrifuge and analysed by western blotting.

Q3. The proposed model in Figure 6f seems incomplete. Does Tip60 co-localize with acetylated lipin 1 on ER membrane? What happens after acetylated Lipin 1 translocate to ER membrane? Sirt1 binds to it to deacetylate it and then recycle it to cytosol? At which step is lipin 1 phosphorylated, before or after Sirt1 binding? The 2KR mutant has lower binding affinity to Tip60 (Figure 6a). It seems phosphorylation is an important component for lipin 1 to complete the cycling.

Thank you for your comment. Since we did not detect significant amount of Tip60 in the ER fraction (Figure R5), and Tip60 did not co-localize with ER marker protein Calnexin regardless of the presence of OA (Figure R6), it is most likely that acetylation of lipin 1 by Tip60 happens in the cytosol and Tip60 doesn't trans-localized to ER membrane after OA treatment. After acetylated lipin 1 translocate to ER membranes, lipin 1 catalyzed the conversion of PA to DAG, feeding into the synthesis of TAG. Similar to Tip60, we did not detect significant amount of Sirt1 in the ER fraction (Figure R5), suggesting that the binding between lipin 1 and Sirt1 likely also occurs in cytosol. Moreover, as shown in Fig. 6c in our manuscript, Sirt1 displayed decreased affinity for phosphorylation-defective lipin 1, indicating that Sirt1 probably favors to bind

phosphorylated lipin 1. Taking into account of the membrane localization of activated mTOR, the kinase for lipin 1 phosphorylation (*Laplante and Sabatini, Cell, 2012*), we speculate that lipin 1 is phosphorylated before Sirt1 binding. With the phosphorylation of lipin 1 by mTOR, followed by its binding with Sirt1 in the cytosol, lipin 1 was reset back to an “inactive” state. In all, we totally agree with you that phosphorylation could be an important event for lipin 1 to complete the cycling.

Figure R5. Treatment of oleic acid (OA) didn’t affect the localization of Tip60 and Sirt1. Western blotting of proteins in subcellular fractions of 3T3-L1 adipocytes with or without OA treatment for 2 h. TCL, total cell lysate; Lamin B, nucleus marker; Calnexin, microsomal marker; β -tubulin, cytosol marker.

Figure R6. Subcellular localization of Tip60 or Sirt1 in 3T3-L1 adipocytes with or without OA treatment. 3T3-L1 adipocytes were treated with BSA or OA for 2 h and analysed by immunofluorescence.

Q4. Does Tip60 S86A mutation cause lower binding affinity to Lipin 1?

Yes. By performing a co-immunoprecipitation assay, we found that, compared to WT-Tip60, S86A-Tip60 mutant demonstrated decreased binding affinity for lipin 1 (Figure R7 and Supplementary Fig. 3b), which may further contributes to its diminished ability in acetylating lipin 1. We have also added this point in our revised manuscript (Page 9, line 3).

Figure R7. S86A-Tip60 mutant demonstrates attenuated binding affinity with lipin 1. HEK293T cells were transfected with Flag tagged lipin 1 and Myc tagged WT or S86A mutant of Tip60, immunoprecipitated with antibody to Flag and immunoblotted as indicated.

Q5. Is the plasma fatty acid level lower in SA mice than in WT mice due to the deficiency of TAG synthesis?

Following your comments, we detected the levels of non-esterified fatty acids (NEFA) in the plasma of WT and *Tip60^{SA/SA}* mice under both fed and overnight fasted conditions. Despite that similar level of NEFA in WT and *Tip60^{SA/SA}* mice was detected under fed condition, the level of NEFA was significantly lower in *Tip60^{SA/SA}* mice than that in WT mice under overnight fasted condition (Figure R8), which is in line with decreased TAG synthesis and adipose tissue in the knockin mice.

Figure R8. Lower level of plasma fatty acid in *Tip60^{SA/SA}* mice than that in WT mice under overnight fasted condition. Plasma levels of NEFA in WT and *Tip60^{SA/SA}* male mice under fed or overnight fasted condition ($n = 10$ for each group). Error bars denote SEM. Statistical analysis was performed by ANOVA followed by Tukey (** $P < 0.01$).

Minor:

Page 8, line 10: “if” should be “whether”.

We have changed this following your instruction (Page 8, line 12).

We thank you again for your constructive comments and patience in reading our manuscript. We hope that the above point-by-point responses have addressed all of your concerns.

Reviewer #3:

Major findings:

*The authors constructed knock-in mice expressing a Tip60 allele that encodes for alanine instead of serine at position 80. This precludes phosphorylation by GSK3 (and perhaps other kinases) on serine 80. This decreases activation of Tip60 in response to this phosphorylation (described in earlier work). These mice were resistant to adipose expansion and weight gain with concomitant maintenance of insulin sensitivity when placed on a high fat diet. Milk triglyceride production was decreased. A mechanistic explanation was supported by evidence that Tip60 directly interacts with and may acetylate the phosphatidic acid phosphatase, lipin1. Site directed mutagenesis identified two lysine residues in lipin1 that were the targets of Tip60-mediated acetylation. Sirt1 was identified as the deacetylase that may reverse the Tip60-mediated acetylation of lipin1. Fluorescence microscopy supported that lipin1 translocates to the ER upon acetylation. Unphosphorylated lipin1 was a better substrate for Tip60, suggesting that dephosphorylation of lipin 1 by an unnamed phosphatase precedes Tip60 acetylation and subsequent activation of lipin1. Parallel studies in *S. cerevisiae* supported that a homologous regulatory system, via acetylation of the lipin1 homolog, Pah1, exists.*

This study provides a physiologically novel mechanism for acetylation regulating phosphatidic acid phosphatase activity and thus, triglyceride synthesis. The experiments presented were exhaustive and thorough with unambiguous, high-quality data throughout. Elucidating novel mechanisms for regulation energy allocation, particularly related to fatty acid allocation and triglyceride synthesis, have clear importance in both basic science and applied disciplines.

We thank you for comments that our work is novel, the experiments are exhaustive and thorough, and that our work has clear importance in both basic science and applied disciplines.

Major Point

1. The discussion would be improved by addressing additional considerations of the data presented. These considerations include addressing the following questions:

a) Nem1p - Spo7p have been implicated in the dephosphorylation and subsequent activation of Pah1p (Pascaul, F. 2014). Are there human homologs of these that might be implicated in the pre-acetylation, phosphorylation-mediated regulation of lipin1?

As you have rightfully pointed out, it has in fact been reported that human homologs of Nem1p-Spo7p, CTDNEP1 and NEP1-R1, play a similar role in regulation the dephosphorylation of lipin 1 (Han et al., *J Biol Chem.* 2012). To further investigate their functions in affecting lipin 1 acetylation, we co-expressed CTDNEP1 and NEP1-R1 in combination with Tip60 and lipin 1 in HEK293T cells. In line with previously reports (Han et al., *J Biol Chem.* 2012), co-expression of CTDNEP1 and NEP1-R1 with lipin 1 strongly induced the dephosphorylation of lipin 1 (Figure R1 and Supplementary Fig. 4e). Meanwhile, the acetylation of lipin 1 was drastically elevated with CTDNEP1 and NEP1-R1 co-expression, which supports our current model that dephosphorylation of lipin

1 facilitates its acetylation by Tip60. We have also added this point in our revised manuscript (Page 12, line 19).

Figure R1. Expression of mammalian homologs of Nem1p-Spo7p, CTDNEP1 and NEP1-R1, leads to lipin 1 dephosphorylation and promotes its acetylation by Tip60. HEK293T cells were transfected with lipin 1, Tip60, CTDNEP1 and/or NEP1-R1, immunoprecipitated with antibody to Flag and immunoblotted as indicated.

b) The authors should acknowledge the presence of a second PAP in the yeast genome, APP1 (Chae, M, 2012), and how App1p activity may account for the TAG synthesis when Pah1p is not active due to diminished acetylation.

Following your instruction, we have discussed the possible role of App1p in TAG synthesis in our current manuscript (Page 16, line 6).

c) To extent have heterozygous, Tip60 wt/SA mice been investigated for these phenotypes studied, is there any evidence that heterozygosity confers intermediate phenotypes?

We have investigated multiple phenotypes of *Tip60*^{wt/SA} heterozygous mice, in parallel with the characterization of wild-type and *Tip60*^{SA/SA} mice, including the body weight, adiposity, liver TAG, plasma TAG/glucose/insulin, GTT and ITT. It was found that *Tip60*^{wt/SA} mice demonstrated similar phenotypes to wild-type mice in all the experiments performed (Figure 1b and data not shown), which implies that the S86 phosphorylation of Tip60 is haplo-sufficient in regulating lipid metabolism.

d) What is a possible mechanism of how oleate / oleic acid influences Tip60 activity?

Actually, in our current model, we found that oleic acid does not directly affect the activity of Tip60, as we did not detect any change of S86 phosphorylation of Tip60 after oleic acid treatment (Fig. 3d and 6e in our manuscript). Instead, it is more likely that oleic acid induces the dephosphorylation of lipin 1, which leads to an increased lipin 1 affinity for Tip60, so as to promote the acetylation and subsequent translocation of lipin 1.

e) Does oleate-induced Tip60 activity also likely change chromatin modification?

In order to clarify whether oleate or *Tip60*^{SA} knock-in may affect chromatin modification, we analysed the acetylation statuses of histone 2A, histone 2B, histone 3, which have been reported to be substrates of Tip60 (Dong *et al.*, *Mol Cell Biol.* 2017), in adipocytes derived from WT and *Tip60*^{SA/SA} MEFs with or without oleate treatment. It was found that neither oleate treatment nor *Tip60*^{SA} knockin affects these chromatin modifications tested (Figure R2).

Figure R2. Neither oleate treatment nor *Tip60*^{SA} knockin affects acetylation of histone H2A, H2B and H3. Adipocytes derived from WT and *Tip60*^{SA/SA} MEFs were treated with or without oleate (OA) for 2 h, and analysed by western blotting.

f) Why was only one lysine (425) conserved with the acetylation target in yeast *Pah1p*? Do the locations of the acetylated lysines yield insight to the structure of *lipin1* and how it binds to the ER?

For the question that why only one lysine (425) is conserved with the acetylation target in yeast *Pah1p*, it is possible that although the general acetylation event of Tip60 is conserved back to yeast, mammalian-specific regulatory mechanisms involved with the other acetylation site (K595) may exist. For example, it has been reported that K595 of lipin 1 can also be sumoylated and potentially regulate nuclear localization of lipin 1 (Liu & Gerace, *PLoS One.* 2009). As for the locations of the acetylated lysines, although the crystal structure of lipin 1 has not been determined yet, several highly conserved domains have been reported within lipin 1 protein, including the NH₂-terminal NLIP domain and the COOH-terminal CLIP domain (Harris and Finck, *Trends Endocrinol Metab.* 2011). The CLIP domain contains the haloacid dehalogenase (HAD) catalytic site required for PAP activity, while a polybasic domain (PBD) near the NLIP domain is the PA-binding motif in mammalian lipin 1. We found that although K425 and K595 are localized in neither NLIP nor CLIP domain, based on disorder prediction using MobiDB, they both reside in structurally flexible regions (aa421-456 and aa566-616), the structural properties of which have been reported to be significantly changed upon post-translational modifications (Bah & Forman-Kay, *J Biol Chem.* 2016). It is thus possible that the acetylation of lipin 1 on these sites could induce disorder-to-order transition to reshape lipin 1 into an ER-binding structure or a conformation that could recruit yet unidentified factor(s) that mediates lipin 1 translocation onto ER. The detailed structural mechanism requires further investigation, which would benefit from our finding of these acetylation sites. We have also added these discussions in our revised manuscript (Page 16, line 9).

Minor points

pg. 2 line23: Glycerol kinase is not commonly considered part of the glycerol 3-phosphate pathway. However, its contribution to proving the pathway with substrate may be a point for regulation and worth mentioning. In that case, glycerol 3 phosphate dehydrogenase should also be mentioned as it is also contributes to the production of glycerol 3 phosphate.

Following your instruction, we have added the related introduction about the contribution of glycerol kinase and glycerol-3-phosphate dehydrogenase to glycerol-3-phosphate TAG synthesis pathway by providing the substrate glycerol-3-phosphate (Page 3, line 1).

pg. 3 line 18: Data / reference should be provided to support that PAP changes subcellular location in response to changing conditions as opposed to being a peripheral protein constitutively present on the ER surface.

Following your instruction, a classic report by *Harris et al (J. Biol. Chem. 282, 277-286, 2007)*, supporting that the PAP changes subcellular location in response to changing conditions, has now been provided in the revised manuscript (Page 15, line 1).

pg. 5 line 5: A nice control would have been a knock-in of a wild-type Tip60 allele so to account for any effects caused by alterations to the Tip60 locus that may have occurred during the insertion of the SA allele. Generating this control would clearly require a good deal of additional effort that would only marginally strengthen the conclusions made.

We totally agree with you that creating an addition mouse strain with a knock-in of a wild-type *Tip60* allele would be a better control for the current *Tip60*^{SA} knockin mice. However, as you have also said, this would require tremendous additional efforts but only marginally strengthen the conclusions made, we just have to leave this work to the future.

pg. 6 line 24: “Lipids, primarily TAG, in the milk supply the majority of the nutrients ...” Consider changing “nutrients” to “calories”

We have now changed this in our revised manuscript (Page 7, line 1).

pg. 13, line 24 The possible molecular / cellular mechanism(s) of the proposed “tissue crosstalk” bear proposing in some detail.

Following your instruction, we have added details on the proposed “tissue crosstalk” (For instance, attenuated lipid synthesis from fatty acids in adipose tissues could lead to complementary increased fatty acid oxidation in metabolic active tissues such as the skeletal muscles in *Tip60*^{SA/SA} mice, that express higher levels of *Cpt1β* (Supplementary Fig. 1m)) in our current manuscript (Page 15, line 12).

Syntax

pg. 2, line 25; That sentence seems to unnecessarily extended by using the semi-colon

We have now corrected this in our revised manuscript (Page 3, line 3).

pg. 4, line 2 “as transcriptional regulators, growing ...” should be “a transcriptional regulator, a growing”

We have now corrected this in our revised manuscript (Page 4, line 5).

pg. 11, line 16: “investigate” to “investigated”

We have now corrected this in our revised manuscript (Page 12, line 13).

pg. 14, line 9 “in turn cooperate with fatty acids to regulate lipin 1 acetylation and TAG synthesis” changing “regulate” to “induce” would make that statement more specific.

We have now corrected this in our revised manuscript (Page 15, line 24).

pg. 43, line 19, 20 “After three cycles of vortexing for 30 s and being left undisturbed for 10 min.” - sentence fragment

We have now corrected this in our revised manuscript (Page 42, line 2).

We thank you again for your constructive comments and patience in reading our manuscript. We hope that the above point-by-point responses have addressed all of your concerns.

REVIEWERS' COMMENTS:

Reviewer #1 (Remarks to the Author):

My major concerns have all been addressed.

Reviewer #2 (Remarks to the Author):

The authors have done a great job to address reviewer's comments. Thus the reviewer suggests to accept this manuscript as is.

Reviewer #3 (Remarks to the Author):

Major points previously raised:

a) The additional experiment performed, shown in supplemental figure 4e nicely addresses the concern expressed. It is curious that upon over-expression of the phosphatase components, the lipin 1 signal detected using the anti-Flag antibody is distinctly reduced (in Suppl. 4e, row 3, column 6). If dephosphorylation decreases lipin 1 stability, that is worth stating. This apparent decrease in lipin 1 protein abundance emphasizes the degree of increased signal of acetylated lipin 1 (in Suppl 4e, row 1, column 6) so it is worth noting. If this observation was not included due to space limitation, that is acceptable.

b) The additional text provided sufficiently addresses this concern.

c) OK

d) OK

e) Mentioning this result seems worthwhile but if it is judged to be ancillary to the hypothesis being tested, leaving out such mention is acceptable.

f) The included text nicely addresses a possible mechanistic basis for the regulation of lipin 1 via acetylation.

Minor points and Syntax issues previously raised.

All of these have been appropriately addressed.

Point-by-Point Responses

Reviewer #1

My major concerns have all been addressed.

We are grateful for the reviewer's agreement that we have addressed the remaining concerns.

Reviewer #2:

The authors have done a great job to address reviewer's comments. Thus the reviewer suggests to accept this manuscript as is.

We are grateful for the reviewer's agreement that we have addressed the remaining concerns.

Reviewer #3:

Major points previously raised:

a) The additional experiment performed, shown in supplemental figure 4e nicely addresses the concern expressed. It is curious that upon over-expression of the phosphatase components, the lipin 1 signal detected using the anti-Flag antibody is distinctly reduced (in Suppl. 4e, row 3, column 6). If dephosphorylation decreases lipin 1 stability, that is worth stating. This apparent decrease in lipin 1 protein abundance emphasizes the degree of increased signal of acetylated lipin 1 (in Suppl 4e, row 1, column 6) so it is worth noting. If this observation was not included due to space limitation, that is acceptable.

We thank the reviewer's expertise comments. Due to space limitation and the fact that investigating the possible link between lipin 1 dephosphorylation and its stability is likely out of the scope of our current study, we thereby prefer not including this info in our current manuscript.

b) The additional text provided sufficiently addresses this concern.

c) OK

d) OK

e) Mentioning this result seems worthwhile but if it is judged to be ancillary to the hypothesis being tested, leaving out such mention is acceptable.

We prefer leaving out such mention as this result is ancillary to the hypothesis being tested.

f) The included text nicely addresses a possible mechanistic basis for the regulation of lipin 1 via acetylation.

Minor points and Syntax issues previously raised.

All of these have been appropriately addressed.

We are grateful for the reviewer's agreement that we have addressed the remaining concerns.